# Limited conservation in cross-species comparison of GLK transcription factor binding suggested wide-spread cistrome divergence

Xiaoyu Tu [1,2,3,8], Sibo Ren[2,8], Wei Shen[2,8], Jianjian Li [2,8], Yuxiang Li[2], Chuanshun Li[3], Yangmeihui Li[3], Zhanxiang Zong[4], Weibo Xie [4], Donald Grierson [5], Zhangjun Fei [6], Jim Giovannoni [6], Pinghua Li [7] ✉ & Silin Zhong [1,2] ✉

Non-coding *cis*-regulatory variants in animal genomes are an important driving force in the evolution of transcription regulation and phenotype diversity. However, cistrome dynamics in plants remain largely underexplored. Here, we compare the binding of GOLDEN2-LIKE (GLK) transcription factors in tomato, tobacco, *Arabidopsis*, maize and rice. Although the function of GLKs is conserved, most of their binding sites are species-specific. Conserved binding sites are often found near photosynthetic genes dependent on GLK for expression, but sites near non-differentially expressed genes in the *glk* mutant are nevertheless under purifying selection. The binding sites' regulatory potential can be predicted by machine learning model using quantitative genome features and TF co-binding information. Our study show that genome cis-variation caused wide-spread TF binding divergence, and most of the TF binding sites are genetically redundant. This poses a major challenge for interpreting the effect of individual sites and highlights the importance of quantitatively measuring TF occupancy.

Transcription factor (TF) binding to *cis*-regulatory elements is crucial to gene regulation. The genome-wide map of all these regulatory interactions is often referred to as the cistrome. Its dynamics enabled species and individuals with similar genes to generate different transcriptional programs and, as a consequence, different phenotypes, and contributed greatly to species' adaptive and phenotypic plasticity[1–3]. With the help of a wide range of high-throughput sequencing techniques, such as SELEX-seq, ATAC-seq, DAP-seq, and ChIP-seq[4–6], it is now possible to map TF binding or identify its footprint genome-wide, which gave rise to a series of cross-species

TF comparisons in yeast and animal models. These studies have shown that homologous TFs with conserved biological functions shared very few binding sites in different species. For example, a pioneer study found that a pseudohyphal development-related TF binds only ~20% of the same target genes in comparison among three *Saccharomyces sensu stricto* species[7]. ChIP-seq of two conserved hepatic bZIP and HB TFs in the liver tissues of five vertebrates (human, mouse, dog, opossum, and chicken) also found that <10% of the binding sites are conserved[8]. The recent large-scale TF analysis conducted by the ENCODE projects also found a low degree of

[1]The South China Botanical Garden, Chinese Academy of Sciences, Guangzhou, China. [2]State Key Laboratory of Agrobiotechnology, School of Life Sciences, The Chinese University of Hong Kong, Hong Kong, China. [3]School of Agriculture and Biology, Joint Center for Single cell Biology/Shanghai Collaborative Innovation Center of Agri-Seeds, Shanghai Jiao Tong University, Shanghai, China. [4]National Key Laboratory of Crop Genetic Improvement, Hubei Hongshan Laboratory, Huazhong Agricultural University, Wuhan, China. [5]Zhejiang Provincial Key Laboratory of Horticultural Plant Integrative Biology, Zhejiang University, Hangzhou, Zhejiang, China. [6]Boyce Thompson Institute, Cornell University, Ithaca, NY, USA. [7]State Key Laboratory of Crop Biology, College of Agronomy, Shandong Agricultural University, Tai'an, Shandong, China. [8]These authors contributed equally: Xiaoyu Tu, Sibo Ren, Wei Shen, Jianjian Li. ✉e-mail: pinghuali@sdau.edu.cn; silin.zhong@cuhk.edu.hk

conservation of TF footprints between humans and mice[9,10], suggesting that the animal cistrome is highly dynamic during evolution.

To measure the impact of TF binding on transcription, exhaustive ChIP-chip experiments have been performed in yeast to identify the binding sites of all known TFs[11–13]. Surprisingly, it was found that many most TF-binding sites have no apparent transcriptional effect. Further studies in higher eukaryotes with large genomes have confirmed that TFs can bind to an unexpectedly large number of sites and most of them appear to have little impact on the nearby gene transcription, suggesting high redundancy and system robustness of the transcription regulatory network[9,14–18]. Interestingly, the conserved TF-binding sites identified in the cross-species comparisons often have the strongest impacts on nearby gene expression[8,19]. In addition, TF-binding sites with strong regulatory potential are often located in super-enhancer regions, which are genome hotspots targeted by multiple co-binding TFs[15]. Therefore, it has been suggested that multiple TFs jointly contribute to the transcription output in a quantitative manner. Individual TF-binding sites may be insufficient to explain transcription, and a cluster of binding sites in the enhancer is key to achieving a precise and robust transcription regulation in the mammalian genome[20].

Despite the wealth of data from animal and yeast models, the modes of cistrome evolution and their relative importance in plants remain underexplored. To study this, one could compare the binding of a well-known TF with conserved biological function in multiple plant species. Photosynthesis is arguably one of the most important and conserved biological processes in plants, and GOLDEN2-LIKE (GLK) TFs are well-known transcription activators controlling chloroplast biogenesis and development[21]. Sub-functionalization of GLKs in monocots such as rice and maize has been reported[22]. They also play a role in fruit development, where GLK1 is switched off, and the GLK2 adapts a latitudinal gradient expression pattern resulting in uneven coloration of the fruit tissue[23,24]. Two redundant copies of *GLK* exist in most diploid angiosperm genomes that have been sequenced to date[25]. Their double loss-of-function mutants in *Arabidopsis*, tomato, and rice showed a pale-green leaf phenotype and down-regulation of photosynthesis genes[21,23,26]. Except for a few well-known target genes, the genome-wide binding profiles of GLKs have yet been determined, which makes it an ideal candidate for a cross-species cistrome comparison.

In this study, we use ChIP-seq to map the GLK-binding sites in five representative plant species and find that most of the GLK-bound genes are species-specific. The conserved GLK-bound genes are often associated with photosynthetic function, and their expression is more susceptible to the GLK mutation. Our results reveal widespread cistrome divergence during plant evolution and the redundant nature of plant TF-binding sites.

## Results

### Genome-wide identification of GLK binding

To study how plant cistrome evolves, we used ChIP-seq to determine the binding sites of GLK1 and GLK2 in the leaf tissue samples from *Arabidopsis* (*A. thaliana*), tobacco (*Nicotiana benthamiana*), rice (*Oryza sativa*), and maize (*Zea may*), as well as the leaf and immature green fruit tissues of tomato (*Solanum lycopersicum*) (Fig. 1 and Supplementary Table 1). These five species were chosen because they could be transformed to express TF fused with an epitope tag and have reference genomes for data analysis. We used the ENCODE2 ChIP-seq processing pipeline with the MACS2-IDR algorithm[27] to process the data, and libraries that passed the QC cut-off value (NSC > 1.05, RSC > 0.9, FRiP > 1% and correlation > 0.8) were used for subsequent analysis (Supplementary Data 1).

GLK is considered as a conserved transcriptional activator controlling photosynthesis and chloroplast development. We first checked some well-known photosynthesis-related genes known to be regulated by the *Arabidopsis* GLKs, such as the chlorophyll-binding proteins *LIGHT HARVESTING COMPLEX A/B* (*LHCA/B*) and those encoding subunits of photosystem I/II. We found that these genes and their homologs in the other four species all have 5′ end proximal GLK1 and GLK2 ChIP-seq peaks (Fig. 2a). They also overlap with the ATAC-seq peaks (Fig. 2a), suggesting that GLKs bind to open chromatin regions in their gene promoter. In total, we identified 960, 1286, 956, 332, and 1089 genes with GLK binding in proximal promoters in *Arabidopsis*, tomato, tobacco, rice, and maize, respectively (Fig. 1, Supplementary Data 2–6).

Previous motif enrichment and protein-binding microarray analyses of *Arabidopsis* GLK have shown that its target genes often contain the GATTCC or RGATTYYY motif upstream of their transcriptional start site[28]. We extracted the sequences of the GLK ChIP-seq peak summit regions and performed motif enrichment analysis. The results

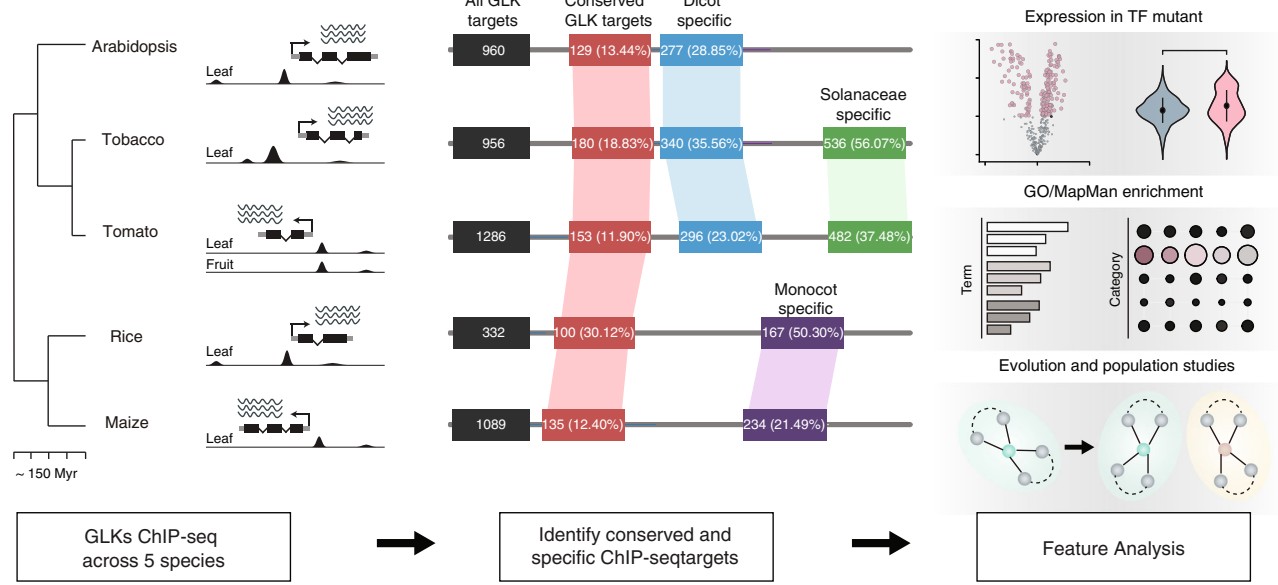

**Fig. 1 | Divergence of GLK TF binding during plant evolution.** ChIP-seq is used to determine the binding of GLK1 and GLK2 in the *Arabidopsis*, tobacco, tomato, rice, and maize genomes. The conserved and specific GLK ChIP-seq target genes are identified, and their features were investigated.

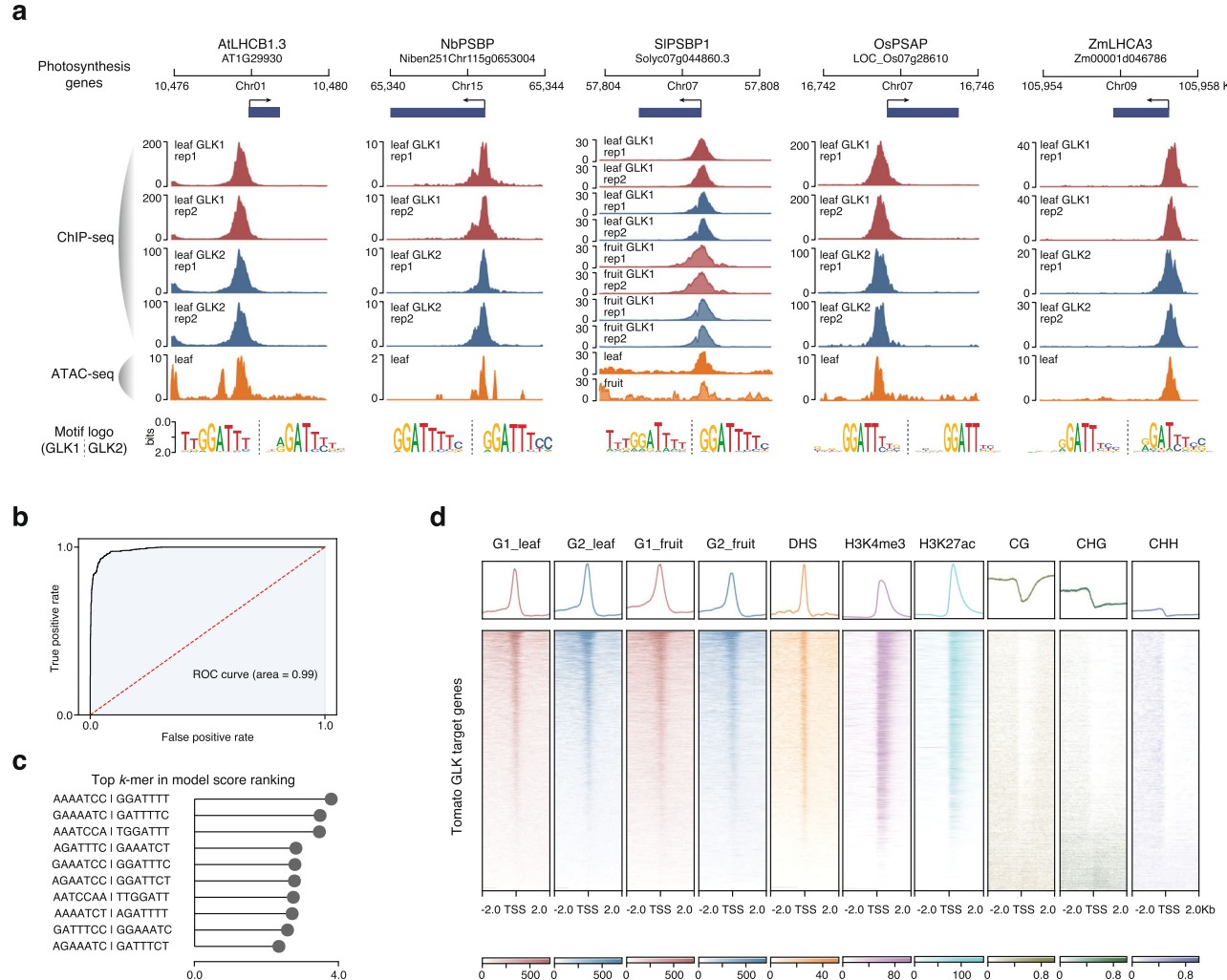

**Fig. 2 | GLK targets identified by ChIP-seq. a** Genome browser tracks showing GLK1 and GLK2 ChIP-seq peaks, as well as open chromatin regions (ATAC-seq) in the photosynthesis gene loci. The motif enrichment results for each GLK are shown below. **b** AtGLK1 bag-of-*k*-mers machine learning model ROC curve. **c** Top 10 scoring *k*-mers in the AtGLK1 model. **d** Heatmap and average signal plot showing the tomato epigenome features near the GLK binding sites. Clustering is performed using the leaf GLK1 ChIP-seq signal. From left to right, GLK1 ChIP-seq in leaf, GLK2 ChIP-seq in leaf, GLK1 ChIP-seq in fruit, GLK2 ChIP-seq in fruit, chromatin accessibility DNase-seq signal, H3K4me3, H3K27ac, DNA methylation at CG, CHG and CHH sites. Regions 2 kb up and downstream of the gene transcriptional start sites are shown. Source data are provided as a Source Data file.

showed that the RGATTYY core motif is indeed enriched in the GLK-binding sites in all five species (Fig. 2a, Supplementary Data 7). It should be noted that the presence of a motif is a necessary, but not sufficient condition for TF binding. Features such as DNA conformation could affect TF binding in vitro, whereas DNA cytosine methylation, histone modifications, chromatin accessibility, and co-binding of other TFs may further complicate TF binding in vivo. Therefore, the number of motifs in a genome often far exceeds the number of actual binding sites[29]. In the case of GLK, there are over two million GLK motif matches (HOMER score > 6) and four hundred thousand GATTCC perfect matches in the maize genome. To overcome the limitation of motif search, machine learning models based on natural language processing have been developed to distinguish the TF bound and unbound regions[30]. One advantage of these models is that it is an ensemble of black boxes that consider multiple short sequences inside the ChIP-seq peaks, which could include motifs of the TF and its co-binding factors or other sequences that influence DNA conformation or epigenome. Such machine-learning tools have already been applied to generate accurate classifier models for 104 maize TFs[6,30].

Hence, we used the *k*-mer grammar tool to train machine-learning classifiers to differentiate DNA sequences in GLK-bound regions and

the background, which are ATAC-seq regions without GLK binding. The result showed that the *k*-mer models can indeed predict GLK binding in five species with high accuracy (Fig. 2b; Supplementary Data 8 and Supplementary Fig. 1). We also extracted the top 10 *k*-mers in these models and found that they often match the GLK core binding motif RGATTYY (Fig. 2c and Supplementary Data 9). For example, the AtGLK1 model has an area under the receiver operating characteristic (ROC) curve of 0.99, an accuracy score of 0.94, and the highest-ranking *k*-mer is GGATTTT (Fig. 2b, c). Such high accuracy suggests that these models could capture both the recognition motif and hidden sequence information in the surrounding region that contribute to TF-binding specificity.

To understand the chromatin environment surrounding the GLK-binding sites, we clustered tomato GLKs ChIP-seq signals centered on the gene transcriptional start sites (TSS) together with different epigenetic features, such as chromatin accessibility, DNA methylation, and histone modifications (Fig. 2d). The GLK-binding regions display typical active chromatin signatures such as high chromatin accessibility and active histone marks H3K4me3 and H3K27ac usually marking promoter and enhancer regions. They also showed low DNA methylation in CG, CHG, and CHH contexts that are often associated with TF-binding sites.

**a**

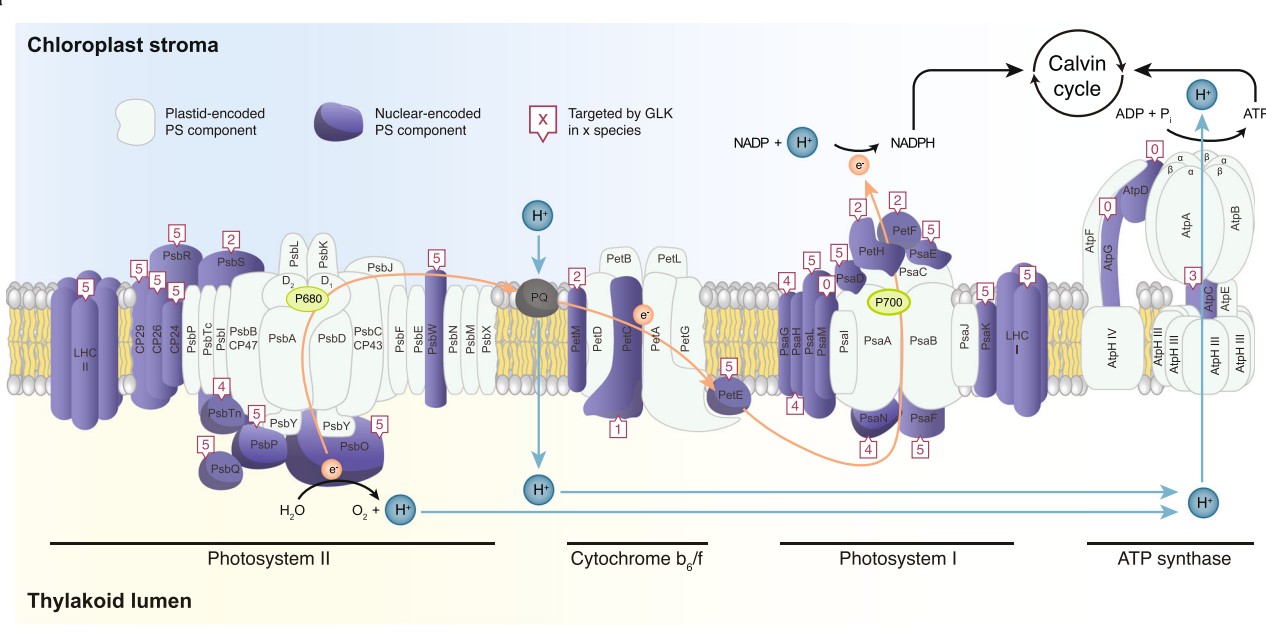

**b**

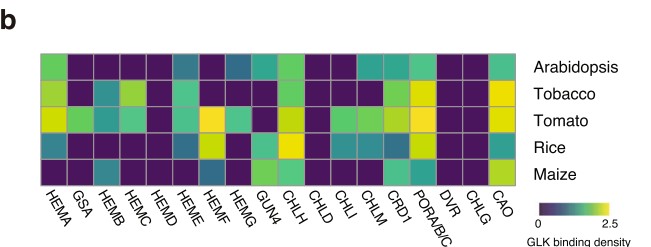

**c**

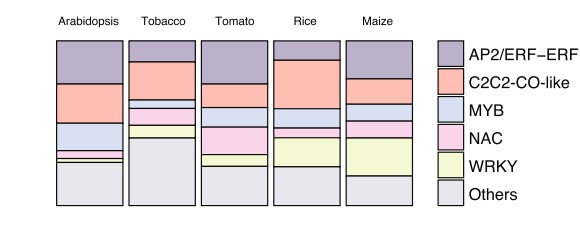

**Fig. 3 | Conserved GLK ChIP-seq target genes. a** Diagram of the photosynthesis electron transfer chain. Components encoded in the nucleus are highlighted in purple. The number inside the boxes indicates the number of species, in which the gene is bounded by GLKs. For the tomato data, only leaf GLK target genes are used. **b** Heatmap showing genes in the chlorophyll biosynthesis pathway and their GLK ChIP-seq signal (−log(foldchange)). **c** Bar chart showing the percentage of conserved TF genes bound by GLK. Source data are provided as a Source Data file.

Previous genetic studies showed that the two GLK genes are functionally redundant[21,23,31]. Consistently, we observed that the GLK1 and GLK2 ChIP-seq peaks overlap in promoters of photosynthetic genes (Fig. 2a). The two GLKs have similar motifs and *k*-mer enrichments (Fig. 2a and Supplementary Data 9). Their ChIP-seq signal heatmap also showed the same read distribution patterns (Fig. 2d), suggesting that the two GLKs have the same binding sites. We then compared the 3472 tomato GLK1 and 6562 GLK2 peaks (IDR cut-off 0.01) and found 2815 overlaps. Even in the 657 GLK1-only peaks that did not pass the GLK2 IDR cut-off, we could still detect GLK2 ChIP-seq signal, and vice versa (Supplementary Fig. 2). To compare the two SlGLKs quantitatively, we also compared their ChIP-seq read coverages in the union of SlGLK1 and SlGLK2 peaks and found that they are correlated. Similar results were also found in all species we examined (Supplementary Fig. 3). Motif enrichment analysis for those GLK1 and GLK2 only peaks showed that they have the GATT core motif (Supplementary Fig. 4), suggesting that the non-overlaps peaks are weaker than the overlapped ones, and we can conclude that the two GLKs have the same binding profile.

**Conserved GLK targets in photosynthesis-related processes**

To understand the function of GLK as a transcription regulator, we examined the GO-term and MAPMAN annotation of its ChIP-seq target genes and performed enrichment analysis. Strikingly, we found that GLKs can bind to most of the nucleus-encoded genes in the chloroplast photosynthetic electron transfer chain, photosystems I and II in particular (Fig. 3a and Supplementary Data 10), as well as genes in the chlorophyll biosynthesis pathways (Fig. 3b and Supplementary Data 11). Besides that, GLK might also play an indirect role in both chloroplast and nucleus transcription, as they consistently bind to pentatricopeptide (PPR) and tetratricopeptide repeat (TPR) protein-coding genes controlling organelle RNA processing[32], as well as CON-STANS/B-BOX TF genes controlling photoperiod in all five species (Fig. 3c and Supplementary Data 2–6)[33,34].

Besides those conserved GLK ChIP-seq target genes, we also found species-specific ones. For example, GLKs only bind to genes encoding PSII subunit S (PsbS) in tomato and rice, while *CHLOROPLAST IMPORT APPARATUS 2* (*CIA2*) is bound by GLKs in *Arabidopsis* and tobacco (Fig. 3a and Supplementary Fig. 5). However, it should be noted that ChIP-seq could identify TF binding sites of a wide range of binding strengths. Such qualitative analysis of ChIP-seq peaks, which label a region as bound or unbound, is known to have limitations[35,36]. For example, weak binding sites are prone to be mischaracterized in a cross-species comparison under different ChIP-seq enrichment and peak caller detection threshold. To compare GLK binding quantitatively between two species, we have calculated the ChIP-seq read counts in the promoter of their homologous gene pairs. The conserved GLK target gene pairs have high ChIP-seq signal in both species, while the species-specific ones only have a high ChIP-seq signal in one species (Supplementary Fig. 6). In summary, our data suggest that although the overall function of GLK is conserved, its binding has diverged during the long evolution time that separated these species.

GO-term enrichment analysis showed that the GLK-bound genes in each species were indeed enriched for photosynthetic terms such as

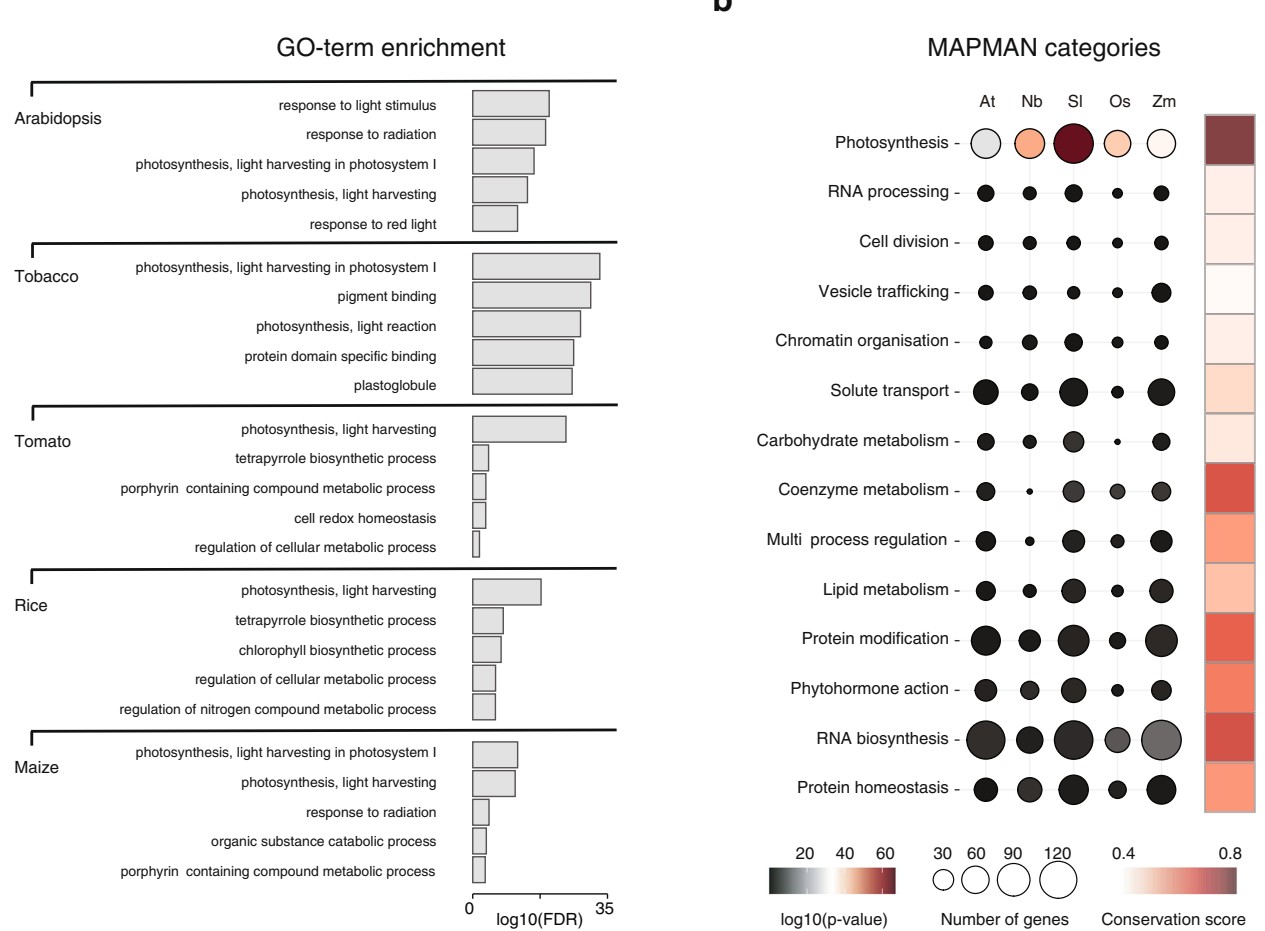

**Fig. 4 | Functional annotation of GLK ChIP-seq target genes. a** GO-term enrichment analysis of the GLK ChIP-seq target genes in each species. **b** MAPMAN functional category enrichment analysis (Fisher's exact, two-sided). The heatmap on the right shows the conservation score for each MAPMAN category. Source data are provided as a Source Data file.

response to light stimulus (GO:0009416) and light harvesting (GO:0009768) (Fig. 4a). We also used the MAPMAN annotation to assign GLK ChIP-seq targets into different functional categories (Fig. 4b). The photosynthesis category is most enriched, even though they only account for ~10% of the ChIP-seq targets in each species. Although more GLK ChIP-seq targets were assigned to other non-photosynthesis categories, none of them were significantly enriched. This could suggest that GLKs bind to photosynthesis-related genes in the ancestral plant, and those binding sites are under strong negative selection, while the non-conserved sites have been gained or lost during evolution.

### Most GLK ChIP-seq target genes are species-specific

Among the five species, the eudicots (*Arabidopsis*, tomato, and tobacco) and the monocots (maize and rice) have diverged ~150 MYA. To study how GLK bindings diverged, we used OrthoFinder to assign all GLK-bound genes in five species into ortholog groups and examined how many common and unique ones could be found in each species. The result showed that very few of them are conserved in all five species. For example, 129, 100, 151, and 205 of the *Arabidopsis* GLK ChIP-seq targets are conserved in 5, 4, 3, and 2 species, respectively. The remaining 375 genes have no ortholog GLK ChIP-seq targets in other species. We also repeated the conservation analysis using gene lists generated by different ENCODE2 peak calling pipelines at various stringencies. If the conservation analysis is affected by false positives or weak binding sites, the conservation rate would increase when a

more stringent cut-off is used. However, when the threshold is raised, the number of identified target genes and the conservation rate remained unaffected or declined, suggesting that our analysis is robust (Supplementary Table 2). Interestingly, if we only count the genes assigned to the photosynthesis category by MAPMAN, their conservation rate is over 90% (Supplementary Table 3), suggesting that the selection pressure on GLK binding is more correlated to gene function rather than the binding strength.

### The conserved binding sites have larger transcriptional regulatory potential and stronger ChIP-seq signal

In animals, it has been shown that the recently gained TF-binding sites are often less important than those conserved ones in terms of their potential to regulate gene expression, despite the TF binding motif in those sites are the same[8]. To test this in plants, we performed RNA-seq for the *GLK1/2* double loss-of-function mutant in *Arabidopsis* and tomato, as well as their wild-type leaves as control (Supplementary Data 12, 13). We identified 1105 differentially expressed genes (DEGs) in *Arabidopsis*, while 266 of them are GLK ChIP-seq targets and 75.94% (202/266) of them are down-regulated in the mutant. We then examined the percentage of DEGs in each conservation group (Fig. 5a). We found that over half (70/129) of the genes in the most conserved group 5 are DEGs, while the non-conserved genes have the least amount of DEGs. Motif-enrichment analysis of the conserved and species-specific binding sites confirmed that they all contain the same RGATTYY motif (Supplementary Figs. 7, 8). This pattern was also

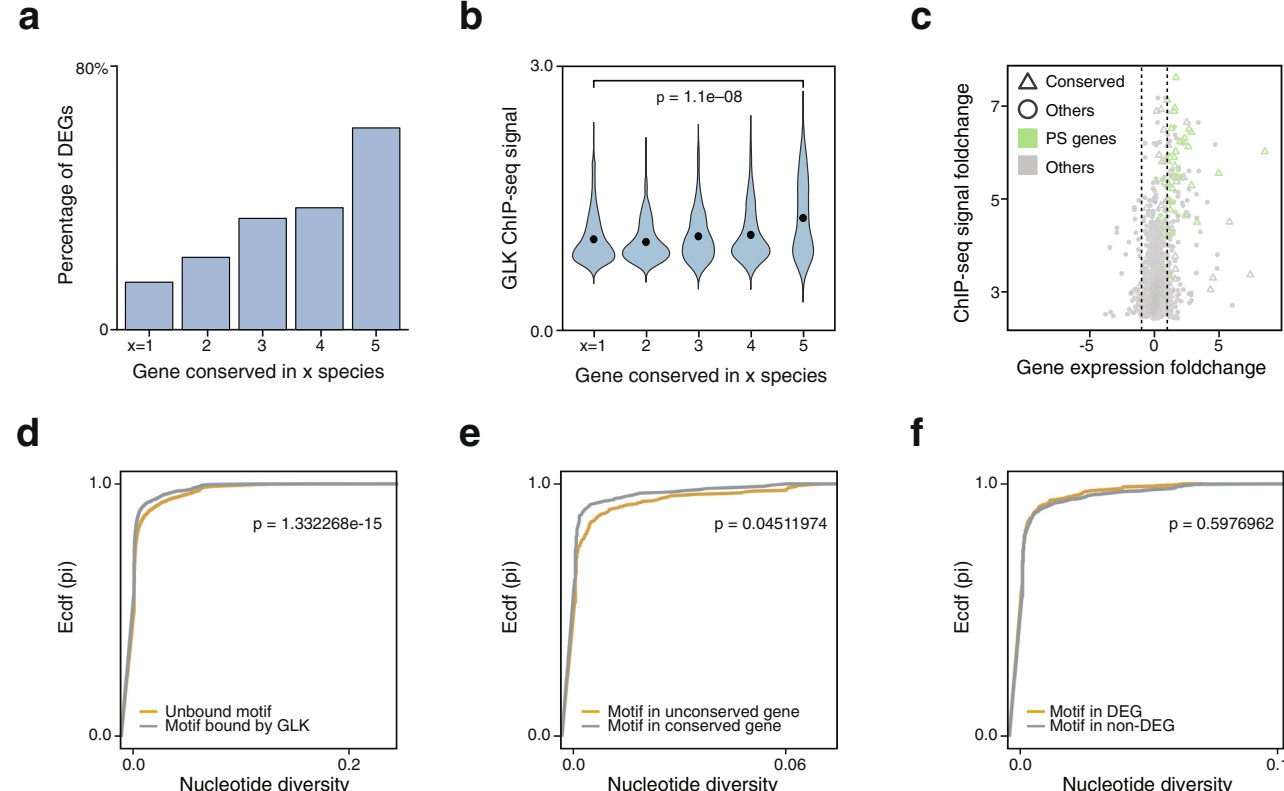

**Fig. 5 | Expression of the conserved GLK ChIP-seq target genes. a** The most conserved *Arabidopsis* GLK ChIP-seq targets genes are often differentially expressed genes in the *GLK* mutants. Target genes are divided into five groups based on their conservation level. **b** GLK binding is stronger in the conserved gene group than in the non-conserved group. The *y*-axis showed the averaged GLK ChIP-seq signal foldchange (log 10) at the peak summit (Kruskal test). **c** Volcano plot shows the log2 fold change of gene expression (WT vs. mutant) and the log2 ChIP-seq signals foldchange. **d–f** Estimator of the cumulative distribution function (Ecdf) plots of nucleotide diversity of GLK motifs (K–S test, one-sided). **d** GLK-bound vs. unbound motifs. **e** Conserved vs. non-conserved motifs. **f** motif in DEGs vs. motif in non-DEGs. Source data are provided as a Source Data file.

observed in tomatoes, suggesting that the conserved plant TF-binding sites also have stronger transcription regulatory potential like the animal ones.

We hypothesize that the non-conserved GLK ChIP-seq target genes recently acquired GLK binding through random sequence variations in the promoter open chromatin regions. Some of those *cis*-regulatory elements have yet to evolve a function that contributes to plant fitness. Without selection pressure, they might be gradually lost. To test this, we imputed TF binding strength based on the average AtGLK1 and AtGLK2 ChIP-seq signal fold-change at the peak summit (Fig. 5b). The result showed that the most conserved binding sites in group 5 indeed have a stronger ChIP-seq signal than the species-specific ones (*p*-value = 1.1E−08). Next, we combined the binding strength and gene expression data and found that the conserved genes such as those annotated with photosynthesis functions often have a stronger ChIP-seq signal, as well as a larger reduction in gene expression in the mutant than the non-conserved ones (Fig. 5c).

**GLK binding sites in non-DEGs are also under negative selection**

Since the conserved GLK ChIP-seq targets are more likely to be differentially expressed in the mutant, could it suggest that the binding sites in non-DEGs or non-conserved target genes are false positive or have no biological function? First, we noticed that over 40% of the conserved ChIP-seq target genes are not DEGs, including some well-known photosynthetic genes (Fig. 5a). For example, the chloroplast ATP synthase subunit gene *ATPC1* has strong GLK binding sites in *Arabidopsis* and tomato, with ChIP-seq signal ranking 70th and 299th, respectively (Supplementary Fig. 9). However, *ATPC1* is only down-

regulated in the tomato, but not in the *Arabidopsis* GLK mutant. In addition, there are also non-DEGs with conserved and strong binding sites. The photoreceptor *CRYPTOCHROME 1* is not differentially expressed in both mutants, but it is a conserved target in all five species with strong ChIP-seq signal (e.g. the 7th strongest in *Arabidopsis*).

We also observed that the binding strength of the conserved sites could change between species. For example, the binding site in PSII subunit gene *PsbTn* is strong in *Arabidopsis* (rank 102nd) and weak in tomato (rank 1083rd), although it is differentially expressed in both (Supplementary Fig. 9). This suggests that the presence of a strong or conserved TF-binding site might not guarantee that the gene would be dependent on the TF for transcription. It is possible that other TFs could compensate for the loss-of-GLK function in a species- and locus-specific manner. Alternatively, TF target genes might be differentially expressed in certain cell types or under specific growth conditions when the TF is knockout. Therefore, it is often difficult to infer the biological function of individual binding sites solely based on limited gene expression data.

To use a different approach to evaluate the potential function of TF-binding sites, one could measure the nucleotide diversity of the site in a given population. If the site is under negative selection, it could suggest that it is important and contribute to the fitness of the species, despite the nearby gene is not differentially expressed in the TF mutant. To test this for GLK, we used the 1001 *Arabidopsis* genome-resequencing data and calculated the nucleotide diversity of its binding sites. We first compared the sites in ChIP-seq peaks against the background, which is the unbound GLK motifs found in open chromatin without GLK ChIP-seq peaks (Fig. 5d). We also compared the conserved against the non-conserved sites (Fig. 5e). As expected, both

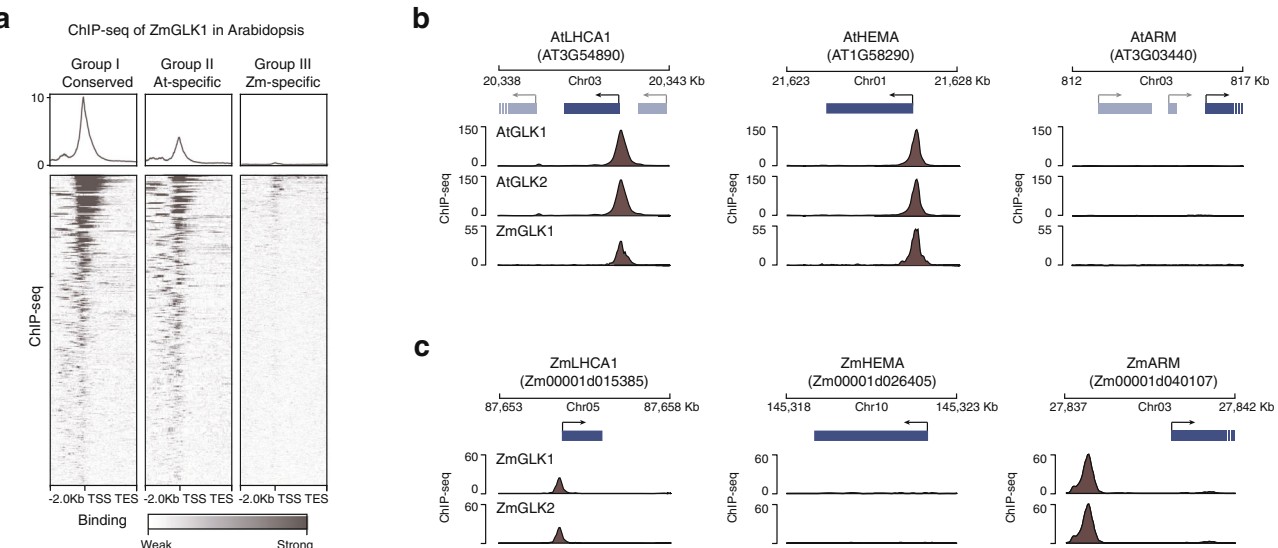

**Fig. 6 | Binding of the heterologously expressed ZmGLK1 in *Arabidopsis* genome. a** Heatmap of ZmGLK1 ChIP-seq signal in three groups of *Arabidopsis* genes. **b** Genome browser tracks showing representative genes in groups one to three. **c** Tracks showing the maize homologous genes.

of their nucleotide diversity scores are lower than the backgrounds, suggesting that they are under negative selection. We then compared the DEGs against the non-DEGs (Fig. 5f), and the result showed that they are not significantly different (Kolmogorov–Smirnov test, $p = 0.597$). This shows that the binding sites in non-DEGs are under similar selection pressure and could contribute to plant fitness in a way that we have not yet understood. Hence, we shall not overgeneralize the relationship between regulatory potential and binding site conservation or binding strength.

**Cistrome dynamics and TF-binding divergence**
Given the large TF-binding divergence, an intriguing question is whether it is caused by the genome sequence variation (*cis*-variation). Alternatively, the evolution of the TF protein (*trans*-variation) could alter its binding preference. To test this, we have transformed *Arabidopsis* with maize GLK1 and performed ChIP-seq to identify its binding sites in a heterologous genome environment.

We first divided the *Arabidopsis* GLK ChIP-seq target and non-target genes into three groups (group I: conserved, group II: *Arabidopsis*-specific and group III: maize-specific) (Fig. 6). The conserved genes in group I, such as *AtLHCA1* and *ZmLHCA1*, are bound by GLK in both species. We found ZmGLK1 ChIP-seq peaks in these *Arabidopsis* gene promoters (Fig. 6b, c). The group II genes do not have a maize ortholog bound by GLK. For example, AtGLKs can bind to the chlorophyll biosynthesis gene *AtHEMA1*, and its ortholog *ZmHEMA* is not a GLK target. But the heterologously expressed maize GLK1 can now bind to its promoter at the same position as the *Arabidopsis* GLKs (Fig. 6b, c). Finally, we examined the maize-specific group III genes, such as the AT3G03440, which encodes a chloroplast ARM repeat protein. When the ZmGLK1 is expressed in *Arabidopsis*, it could no longer bind to its promoter. The ChIP-seq signal heatmaps of this heterologously expressed ZmGLK1 showed that it has recapitulated the *Arabidopsis* GLK's binding pattern in both group I and II genes, but did not bind to the maize-specific genes in group III (Fig. 6a). Taken together, our data showed that TF binding is largely determined in *cis* by DNA sequence variation rather than in *trans*.

**Use of machine learning model to predict transcription outcome**
To better understand why only part of the genes with proximal GLK-binding sites is differentially expressed, we sought to quantitatively compare their genome features such as TF-binding strength, the initial

expression level in wild-type leaf, distance of the binding site to the gene TSS, as well as other TFs' co-binding information inferred from the *Arabidopsis* TF DAP-seq data collection. Unsurprisingly, the differentially expressed genes (DEGs) have stronger ChIP-seq signal, a higher gene expression level in the leaf, and closer GLK-binding sites (Fig. 7a). We also found TF DAP-seq data that has a co-binding pattern correlated with the DEG status (Fig. 7b). However, it must be noted that none of these features on its own is sufficient to predict whether a GLK ChIP-seq target gene would be differentially expressed. Hence, we trained random forest classifier models to consider different quantitative genome features together. To avoid the effects of outliers given the small dataset of 202 DEGs, we randomly sampled the training and test data 500 times to train 500 independent models. The average AUC, ACC, Recall, and F1-score of the models are 0.77, 0.698, 0.685, and 0.688, respectively (Fig. 7c). This showed that combining different classes of genome features could indeed improve our interpretation of the genome regulatory code.

This method also enabled us to test what feature is important for the model's accuracy. We found that removing the TF co-binding data resulted in the largest decrease in the model performance, suggesting that the co-binding TFs could hold vital information for predicting the transcription outcome (Fig. 7d). The random forest model also allowed us to calculate a feature importance score for each co-binding TFs in the DAP-seq data. We found that some of the well-known photosynthesis and light-signaling regulators such as MYB-related, C2H2 and GBF TFs are among the top contributors (Fig. 7e). Although correlation does not imply causality, it is possible that the differentially expressed genes have been co-regulated by GLKs and these TFs in the ancestral plant, and they have evolved into an indispensable part of the gene regulatory network. It is similar to the observation in animal genomes that the TF binding sites with strong regulatory potential are often located in the enhancers bound by a large number of TFs[15].

**Tracking GLK binding divergence after genome duplication**
The tomato genome has experienced the ancient eudicots gamma duplication (~120 MYA) and the recent *Solanum* lineage one (~70 MYA). As the promoter regions bound by the TFs are also duplicated during those events, it provides a unique opportunity to estimate whether the interaction between GLK and its targets occurred before or after one of the duplications. We found that the tomato GLKs bind to a pair of genes encoding *LIGHT HARVESTING COMPLEX A4* (*LHCA4*) in synteny blocks in chromosomes 3 and 6. Both genes

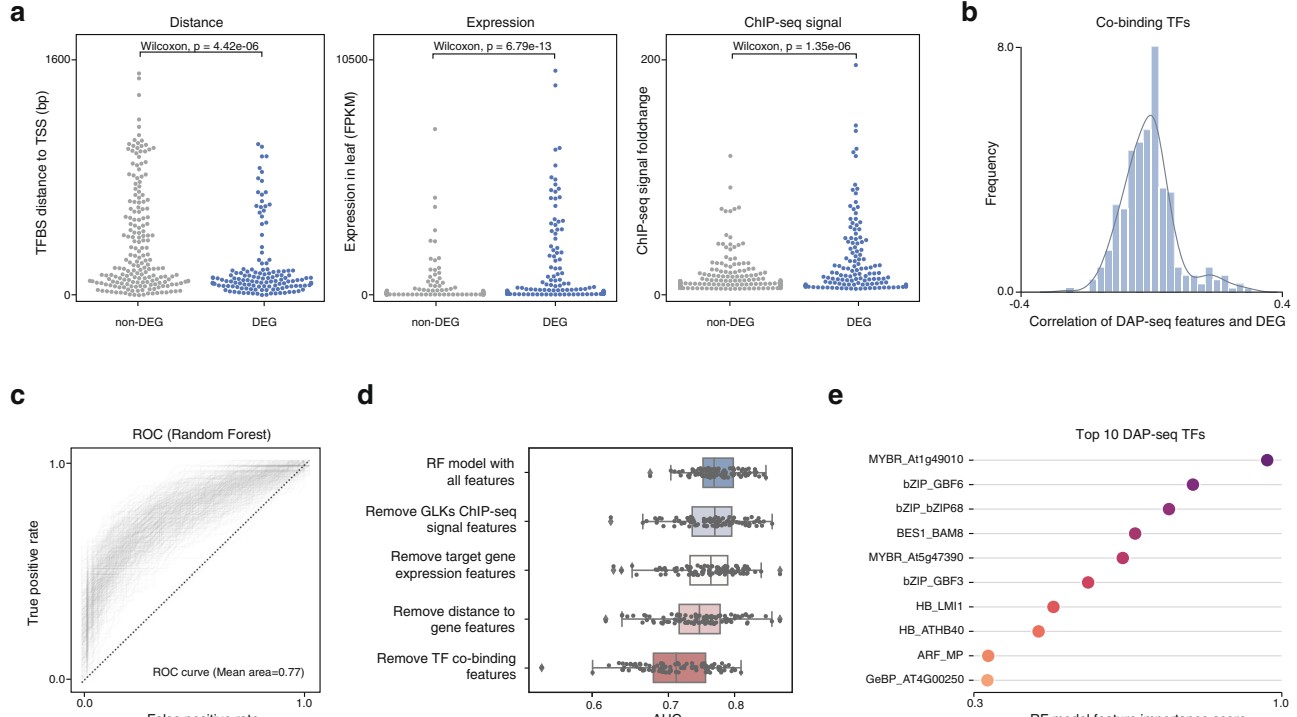

**Fig. 7 | Machine learning model to predict target gene expression. a** Different quantitative genome features of the differentially and non-differentially expressed GLK ChIP-seq target genes in the *Arabidopsis* GLK double mutant (Wilcoxon test, two-sided, *n* = 328). **b** Histogram of Pearson correlation coefficient of DAP-seq TFs co-binding features and the GLK targets' DEG status. **c** ROC curves of the 500 random forest models. **d** Boxplot showing the area under the ROC curve (AUC) of the 500 random forest models after removing different features. The box contains the 25th–75th percentiles of the dataset, the central line denotes the median, and the whisker represents 1.5 IQR. **e** Top 10 DAP-seq TF features of the models, ranked based on their average importance score in 500 random forest models. Source data are provided as a Source Data file.

are down-regulated in the mutant (Fig. 8a). These two synteny blocks have a high synonymous substitution rate (*K*s = 1.689) and are not located in the previously reported sub-genome regions that resulted from the recent genome triplication[37], suggesting that the GLK bindings in *LHCA4* have been preserved after the paleo-duplication for over 100 MYA. However, genes could be lost or rearranged after genome duplication, and as a result, most of the tomato genes are no longer located in synteny blocks. The dupli-cated genes could still be identified as homologous pairs that both are targeted by GLK. We calculated the percentage of homologous GLK ChIP-seq targets in the five conservation groups and found that the most conserved group 5 indeed has more homologous target genes, suggesting that the conserved binding sites are ancient, and the species-specific ones are acquired more recently.

Unlike tomatoes, the maize genome experienced a recent tet-raploidy event 5–12 MYA[38]. Most of its duplicated genes could still be traced back to subgenomes 1 and 2. We then compared the GLK ChIP-seq signal of the duplicated ChIP-seq target genes using the Kendall rank correlation coefficient test. Kendall's tau is 0.174 (*p* = 0.06322), suggesting that the correlation is very weak, and duplicated GLK binding sites have already changed after such a short evolutionary period. We also hypothesize that the rapid change in TF binding might allow the duplicated genes to be expressed in a different tissue, which could further give rise to neofunctionalization. To test this, we calculated the expression correlation of the maize GLK ChIP-seq target gene pairs in the two subgenomes using the maizeGDB's tissue gene expression data. The result showed that the pairs that retained GLK binding have the highest expression correlation while losing GLK binding led to sig-nificantly lower expression correlation, confirming our hypoth-esis (Fig. 8c).

## Discussion

Cross-species comparison of GLK binding showed that plant cistrome dynamics can also cause widespread TF-binding changes like those observed in the yeast and animal genomes[9,14,15,17,18]. Our observation is also in line with previous plant TF ChIP-seq analysis of the SEPALLATA3 MADS-box TFs in *A. thaliana* and *A. lyrata*, which found that less than a quarter of their binding sites are conserved[39]. Despite the fact that the non-conserved and weak TF binding sites are less likely to influence gene expression, it has been suggested that we should not consider them as non-functional[14]. Because eukaryotic genes are often regu-lated by multiple TFs, and the combined input could exceed the threshold of transcription activation, generating redundancy for the whole gene regulatory network. In addition, the regulatory potential of a TF-binding site could not be fully demonstrated by individual tran-scriptome analysis, since a non-differentially expressed TF target gene could become differentially expressed in another growth condition or tissue. Consistently, our data showed that the GLK-binding sites in the non-differentially expressed GLK ChIP-seq target genes are also under similar negative selection as those in the differentially expressed ones. Hence, it is important to study how TFs co-regulate gene expression in order to fully understand the genome-regulatory code.

Many plant TF studies, particularly those in crops that are hard to transform, have often inferred gene function based on mutant com-plementation or over-expression in a model plant like tomato and *Arabidopsis*. Our finding gives reason for caution in interpreting the result of such studies. If TF binding could be influenced by the genome *cis*-regulatory dynamics, one might have missed the species-specific function of a TF when expressed in a foreign genome. For example, ETHYLENE INSENSITIVE3 class TFs are key regulators of the conserved ethylene signaling pathway[40]. We have previously found that some climacteric fruit species have gained EIN3 binding sites in the

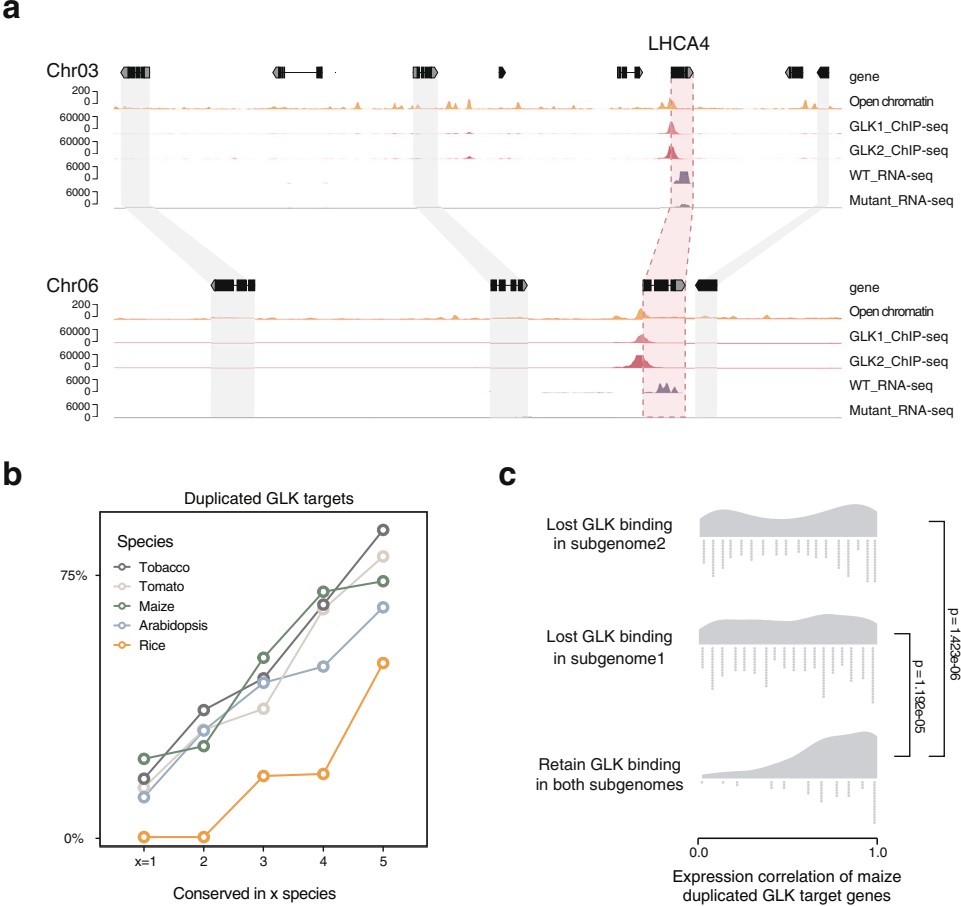

**Fig. 8 | Duplicated GLK ChIP-seq target genes. a** Genome browser tracks showing part of the synteny blocks containing two tomato LHCA4 genes. The open chromatin, GLK ChIP-seq and RNA-seq tracks are shown. **b** The percentage of duplicated GLK ChIP-seq target genes in each conservation group. **c** Pearson correlation of the expression pattern of maize GLK ChIP-seq target gene synteny pairs in subgenomes 1 and 2. Pairs retained both GLK binding sites after genome duplication have the highest correlation (t-test, one-sided). Source data are provided as a Source Data file.

promoters of their ripening genes, which have been evolved into indispensable parts of their positive feedback loops controlling ripening[41]. Thus, although the EIN3 genes in those fruits can complement the *Arabidopsis EIN3* loss-of-function mutant and generate the same phenotype when overexpressed, their recently evolved roles in fruit ripening could not be tested in the *Arabidopsis* genome.

One limitation of our study is that, without a ChIP-grade antibody, we must overexpress the TF fused with an epitope tag to perform ChIP-seq, making it difficult to control for protein level, as well as where and when the TF is expressed. Since ChIP-seq is an enrichment analysis, factors such as the protein abundance and cellular location, the specificity of the antibody, sequencing depth, library complexity, and even small details in the ChIP experiment itself such as input to antibody ratio, sonication, and washing steps could influence the final ChIP enrichment result. Strong TF-binding sites could be easily detected even when the enrichment is poor. But the number of weak TF-binding sites identified could be easily affected by the ChIP-seq enrichment, making cross-species or cross-experiment comparison difficult[35]. For example, we have previously shown that the sensitive ChIPmentation assay could detect five times more binding sites with the same antibody against the tomato MADS-box TF RIN than the traditional ChIP-seq method[41,42]. With the development of CUT&Tag and single-cell ChIP-seq, it is likely that the sensitivity could be further increased. In addition, the next generation of peak calling algorithms based on neural network models or an ensemble of different methods is now able to find weak TF-binding sites from noisy ChIP-seq data[43]. The increased sensitivity means

more weak binding sites could be discovered in the future, posing a major challenge for data analysis.

Another intriguing question is why the genome would retain so many genetically redundant TF binding sites without a strong regulatory role. It has been hypothesized that gaining regulatory complexity is the key to the evolution of multicellular organisms with complex cell types, by enabling the common genes to be exploited multiple times to generate different temporal and spatial transcriptional programs and biological outcomes[44]. As the cistrome evolved rapidly, *cis*-regulatory elements are generated or disrupted by sequence variations, which leads to the recruitment of new TF or the weakening of existing TF binding, and as a result, different transcriptional outcomes and phenotypes could be generated. Therefore, the observed divergence of GLK binding across plant species, as well as its conserved core regulatory interactions could be a real-time snapshot of evolution in action.

Several studies in yeast and animal models have shown that even the weak TF-binding sites could play an important role in evolving new gene regulatory programs and are required for achieving high TF-binding specificities[45,46]. For example, replacing the native low-affinity Hox-binding site with an artificial high-affinity one caused ectopic gene expression in *Drosophila* embryo[47]. Recent plant TF studies using high-sensitivity ChIP-seq have shown that each TF can bind to around 10,000 loci[5,48]. The *Arabidopsis* cistrome project found ~2.7 million in vitro binding sites for 529 TFs. If a TF can bind to a third of the genes in the genome, it is unlikely that all binding sites would be equal in terms of regulating transcription. Therefore, one might need to

consider the TF binding quantitatively and cooperatively, as we demonstrated in using the random forest model to predict GLK ChIP-seq target gene expression. In addition, a recent analysis of embryonic stem cell reprogramming factors Oct4, Sox2, Nanog, and Klf4 revealed that the spacing and direction of TF binding can also encode critical regulatory information[49], while similar analysis for plant TFs has largely fallen behind.

## Methods

### Plant materials

The *Arabidopsis glk1/glk2* mutant[25] was obtained from the European *Arabidopsis* Stock Centre (N9807). Transgenic *Arabidopsis* plants were grown on Murashige and Skoog (MS) salts with 25 μg/mL hygromycin B, and 0.8% (w/v) agar. Plants were grown at 22 °C under long-day conditions (16-h light) in a growth chamber. Rice cultivar Nipponbare was used as the wild type. Rice plants were grown at 28 °C with long-day conditions in the greenhouse. The tomato *glk1/glk2* double mutant generated by CRISRP/Cas9 in microTom background was kindly provided by Prof. Yu Pan, and we have confirmed the deletions by Sanger sequencing (Supplementary Fig. 10). Both tomato and tobacco (*Nicotiana benthamiana*) were grown in a growth chamber with a 12 h-light (25 °C) and 12 h-dark (20 °C) cycle.

For the *Arabidopsis* GLKs, we fused their cDNAs with the C-terminal GFP tag and transformed them into the *glk1/glk2* double mutant for ChIP-seq (Supplementary Table 1). Transgene expression of tomato *GLK2* is known to cause co-suppression[23]. Hence, we transformed the SlGLK2-GFP construct into the tomato cultivar Micro-Tom, which harbors a loss-of-function mutation in the *GLK2* locus. As the tomato GLKs are also involved in early fruit development, we also performed ChIP-seq using immature fruit at 27 day-post-anthesis. The cDNA of the rice GLKs were fused to an HA tag and transformed into wild-type rice under the control of the ubiquitin promoter. For these stable transgenic plants over-expressing GLKs, we selected the transgenic line with high protein expression levels for ChIP-seq. Tobacco is a recent allotetraploid and has two copies of each GLK with over 90% sequence identity. Hence, we selected the two GLK genes with the highest leaf expression level for the analysis. We used agroinfiltration to transiently express the NbGLKs with an HA tag for ChIP-seq. The maize (*Zea mays*) GLKs ChIP-seq data were obtained from a previous study[6]. The maize GLK1-HA construct was also transformed into the *Arabidopsis glk* double mutant for ChIP-seq.

### ChIP-seq

Fully expanded leaves from *Arabidopsis*, rice, tomato, and *Nicotiana benthamiana*, as well as pericarp samples from immature green tomato fruits at 27 DPA, were harvested and cross-linked with 1% (w/v) formaldehyde. The nuclei were then isolated by filtration and centrifugation, and the chromatin was sonicated to sub-kb fragments using Bioruptor. Anti-GFP (#A-11122, ThermoFisher) and anti-HA (#C29F4, Cell Signaling Technology) antibodies were used for ChIP. The protein A/G Dynabeads (10 μL) were incubated with 2 μL antibody for 1–2 h in low salt buffer (20 mM Tris–HCl pH 8.0, 150 mM NaCl, 2 mM EDTA, 0.1% Triton X-100, 0.1% BSA). The beads were then washed with low salt washing buffer (20 mM Tris–HCl pH 8.0, 150 mM NaCl, 2 mM EDTA, 1% Triton X-100, 0.1% SDS) before adding to the sonicated chromatins, were incubated overnight with the Dynabeads with antibody, and washed twice with low salt washing buffer (20 mM Tris–HCl pH 8.0, 150 mM NaCl, 2 mM EDTA, 1% Triton X-100, 0.1% SDS), high salt buffer (20 mM Tris–HCl pH 8.0, 500 mM NaCl, 2 mM EDTA, 1% Triton X-100, 0.1% SDS), LiCl buffer (20 mM Tris–HCl pH 8.0, 250 mM LiCl, 1% Triton X-100, 0.7% sodium deoxycholate, 1 mM EDTA) and TE. The immunoprecipitated DNA was tagged on-bead with TS-Tn5 for 45 min. The beads were then washed twice with high salt buffer and TE. Samples were reverse crosslinked overnight at 65 °C. DNA was purified with Qiagen MiniElute and PCR amplified. The libraries were sequenced with a Hiseq X 150 bp paired-end read mode.

### RNA-seq

Total RNA was extracted with RNeasy Mini Kit (Qiagen). Messenger RNAs were isolated with Oligo d(T)25 magnetic beads (New England Biolabs) and used for Illumina TruSeq library preparation. The libraries were sequenced on Hiseq X with 150 bp paired-end mode.

### ChIP-seq data analyses

ChIP-seq reads were aligned to plant reference genomes (*Arabidopsis thaliana* TAIR10, *Nicotiana benthamiana* version 2.5.1, *Solanum lycopersicum* SL4.0, *Oryza sativa* MSU7.0, and *Zea mays* RefGen_v4) using Bowtie 2 (version 2.3.2). For paired-end, 150 bp reads, the 100 bp from 3' end were trimmed with the bowtie2 parameter −3 100. Unmapped and low mapping quality reads were filtered with SAMtools (version 1.9) using the parameters "-F 4 -q 20". Duplicated reads were removed with the subcommand "rmdup" in SAMtools. Pearson correlation coefficients were calculated with the command "multiBigwigSummary" in deepTools to evaluate the reproducibility of the biological replicates. Only biological replicates with a correlation coefficient > 0.8 were kept for further analyses. The ENCODE2 ChIP-seq pipeline with MACS2 (version 2.2.1) and IDR (version 2.0.4.2) were used for peak calling. Peak summit positions were retrieved by using the "−call-summits" function in MACS2. The regions ±75 bp from the summit position with signal fold-change in NarrowPeak format were then supplied to IDR, and the regions passed the IDR 0.01 and summit signal foldchange cut-off were kept and resized back to 150 bp. They were then associated with genes based on summit distance to the TSS. Depending on genome size, the ChIP-seq peak summit overlaps with the putative promoter region 1.5, 2.5, 2, 1.5, and 2.5 kb upstream of the TSS were associated with genes in *Arabidopsis*, tobacco, tomato, rice, and maize, respectively. We also used the ENCODE2 spp-IDR-TIP pipeline from PhantomPeakQualTools (version 1.14) at two different cut-offs ($p = 0.05$ and $p = 0.01$) for comparison.

### RNA-seq analysis

RNA-seq reads were first mapped to rRNAs, and the clean reads were then mapped to the corresponding plant genomes by HISAT2 (version 2.1.0). Read counts were calculated with HTSeq (version 0.11.0). Differential gene expression analysis was performed using DESeq2.

### Function and motif enrichment analysis

Functional GO enrichment analysis was performed by a web-based toolkit for the agricultural community agriGO v2.0 (http://systemsbiology.cau.edu.cn/agriGOv2/). Gene functional category enrichments were performed using MAPMAN. The target genes in the five species were compared using OrthoFinder (version 2.2.7) to determine the homologous relationship based on protein sequence similarity. We used a conservation score to quantify how conserved a group of genes (e.g. the eudicot shared GLK target genes in one MAPMAN category) based on the events (observed) and the pair-wise comparisons performed (expected). De novo motif discovery and motif search were performed using HOMER (http://homer.ucsd.edu/homer/motif/). For the nucleotide diversity analysis, we first identified the GLK motif position in the *Arabidopsis* GLK ChIP-seq peaks using HOMER (cut-off: motif match score > 6). We also identified the unbound GLK motif hits in the open chromatin regions (ATAC-seq peaks) without GLK ChIP-seq signal as background. The nucleotide diversity score of each motif position is retrieved from the variant collections identified by the 1001 *Arabidopsis* genome project (https://1001genomes.org/data/GMI-MPI/releases/v3.1/).

## Machine learning analyses

We used the *k*-mer grammar tool (https://bitbucket.org/bucklerlab/k-mer_grammar/src/master/) to generate machine-learning models to distinguish sequences of GLK-binding sites from sequences in open chromatin regions without GLK ChIP-seq peaks. For GLK-binding sites, we tested both the 150 and 300 bp sequences centered at the ChIP-seq peak summit. The sequences of the same length centered at the ATAC-seq peak summit are used as the background, and open chromatin overlaps with weak ChIP-seq peaks (IDR 0.05) were removed. As there are often 10 times more open chromatin regions than ChIP-seq peaks, an equal number of open chromatin regions are randomly selected to generate a balanced dataset for training the *k*-mer models. The balanced input datasets were then randomly split into 80% training and 20% test sets for the *k*-mer grammar tool.

To study the different transcriptional regulatory functions of GLK-binding sites, we also used a random forest classifier to discriminate down-regulated DEGs from non-DEGs. Features used for RF model training are the GLK1 and GLK2 ChIP-seq signal foldchange values at the genes' closest ChIP-seq summit, the distance from the summit to the genes' transcriptional start site in base pair, the genes' initial expression level in wild-type leaf in FPKM, and *Arabidopsis* TF DAP-seq peaks obtained from the plant cistrome database (http://neomorph.salk.edu/dap_web/pages/index.php). GLK target genes with DAP-seq peaks overlapping with the 150 bp GLK ChIP-seq summit regions were used. DAP-seq data were converted to categorical features (1 bound, 0 unbound) for training. The DAP-seq TF datasets with low features-label correlation (<0.1) were discarded. As there are more non-DEGs than DEGs, we randomly selected an equal number of the non-DEG labels to match the DEGs to obtain a balanced training dataset, and 80% of the data were used for training while the remaining 20% were used for the test. Given the small sample size, we also randomly re-select the training and test data 500 times to generate 500 models to calculate the average AUC and model scores.

## Synteny analysis and *K*s calculation

Detection of synteny and collinearity was performed by MCScanX with default parameters. The synonymous substitution rate (*K*s) between homologs was computed by KaKs_Calculator (version 2.0).

## Reporting summary

Further information on research design is available in the Nature Portfolio Reporting Summary linked to this article.

## Data availability

The data generated in this study have been deposited in the NCBI database under accession code PRJNA682315 GSE220115 or GitHub [https://github.com/rensabella/GLK-project/tree/main/data]. Published data used in the study can be accessed at PRJNA518749 and PRJNA743574. The *Arabidopsis* TF co-binding data is available in the plant cistrome database [http://neomorph.salk.edu/dap_web/pages/index.php]. The *Arabidopsis* nucleotide diversity data is available in the 1001 Genomes database [https://1001genomes.org/data/GMI-MPI/releases/v3.1/]. The maize 23 tissues' gene expression data is available at maizeGDB [https://www.maizegdb.org/expression]. Source data are provided with this paper.

## Code availability

All codes have been deposited in GitHub [https://github.com/rensabella/GLK-project].

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

## Acknowledgements

This work is supported by funding from the NSFC (32100438) (to X.T.); South China Botanical Garden CAS, Hong Kong GRF 14109420, AoE/M-403/16 and the State Key Laboratory of Agrobiotechnology (to S.Z.); The National Key Research and Development Program of China (2022YFF1001700) and NSFC (31871313) (to P.L.). We thank Dr. Fei Lu, Dr. Yijing Zhang, and Dr. Peng Wang for their helpful discussion, as well as Dr. Yu Pan for providing plant materials.

## Author contributions

S. Z. and P.L. designed the project; J.L., Y.L., and X.T. performed the experiments; X.T., R.S., W.S., C.L., Z.Z., and W.X. performed the computational analysis; S.Z., X.T., G.D., Z.F., P.L., and J.G. wrote the paper. All the authors read the paper and agree with the final version.

## Competing interests

The authors declare no competing interests.
