## [Peer Review File · Nature Communications]

Limited conservation in cross-species comparison of GLK transcription factor binding suggested wide-spread cistrome divergenceReviewers' Comments:

Reviewer #1:

Remarks to the Author:

This manuscript deals with a comparison of GLK transcription factor (TF) targets in different flowering plant species. The topic is very interesting : understanding the evolution of TF targets is fascinating and timely topic and reports are still rare (doi: 10.1093/molbev/msv210 should be mentioned). It focuses on the GLK TF that has a series of well established targets in photosynthesis genes as well as additional newly evolved specialized functions.

Overall, the message of the paper is convincingly supported and well illustrated. The GLK TFs share only a subset of conserved targets and they are mostly present within conserved binding. The binding conservation not associated with regulation is quite intriguing and could be commented and examined closer. Some of the conclusions of this work might appear almost trivial as they are logic but I truly believe that such quantitative study where the targets have been examined experimentally combining binding (ChIP) and regulation (RNA-seq in mutants) is essential so that rules can be established and not just anticipated. To me, this is the main significance of this work that otherwise does not really involve any scientific or technological breakthrough.

Here are a series of comments

Major points:

1. I am surprised that GLK binding sites models have not been to identify whether the binding could be predicted without performing ChIP or to test whether conserved and less conserved targets have different types of binding sites
2. The introduction states that some new roles have been acquired in tomato but then the paper focuses on conserved targets and much less on newly gained targets (bound and regulated). It would be nice to examine whether some true targets specific to one or a few related species lead to some new functions.

Minor points and suggestions

1. L1 and 32: "neutral" seems inappropriate. I understand that "neutral" is used here because of all the non-conserved targets but I am unsure "non-conserved" equates "neutral". Among the specific targets, many could have evolved under positive selection but this is not tested (and probably not testable with the data at hand?).
2. In the title "reprogramming" is somewhat obscure. To me it would mean that a given existing function is achieved via a modification in the regulatory network but this is not demonstrated in this study. Perhaps a fairer title could be: "Limited cistrome conservation in cross-species comparison of transcription factor binding implied/suggest wide-spread cistrome reprogramming during plant evolution"
3. Abstract : L27 what is a primary TF ? L28 dispensable for the expression of the non-conserved ones. L30 : explain the gamma duplication
4. The fact that ChIP-seq peaks come from overexpression constructs of GLK genes may inflate the number of non-conserved targets because the TF overexpression may drive binding where it would not occur in a normal plant. What do the authors think about it?
5. In the abstract "Our study reveals the TF binding dynamics in plant species diverged over 100 million years" I am not sure what is meant here, and how it relates with the results. Could you clarify?
6. L 165: I do not understand how 646 targets can be conserved between the 5 species and then only 136 of the tomato targets be conserved in all 5 species. Those 646 should be present or did I miss something ? This point needs to be clearer
7. L39 : I would replace "...species with similar genes to create..." by "...individuals within a species to evolve..." because species already have different phenotype.
8. L66 to 92: a lot of this could be in method

9. L105: "epigenome features" needs to be defined
10. L204: in figure 5b: enrichment is not define. I assume it is FPKM (?)
11. L227 remove "(Figure 5c and d)". L229: add Figure 5d.
12. L243: "are not located in the previously reported sub-genome regions that resulted from the recent triplication" what is this triplication? Didn't it affect the full genome? a reference may be needed here.
13. L279: strong selection -> strong negative selection
14. L291: "has" is missing.
15. L35:, trimmed with which tool and what are these options?
16. Fig. 5a: X value or X=1 but not X1. The term motif is not clear. It is usually used for TFBS.

Reviewer #2:

Remarks to the Author:

This manuscript by Wei Shen et al describes a comprehensive effort to map the targets of GLK transcription factors in five angiosperm species using ChIP-seq. This is quite an achievement, and a more realistic description of the work than the grand title suggests. Once you dig down to the results, I think the title might be over-selling the work somewhat, but we all need to grab attention somehow, so I don't have any real problem there.

I have to be honest, I am ambivalent about this manuscript. There's quite a lot of work in it, and it is well presented, but I feel it is written as a bit of a "just-so story". It seems like the authors set out to compare the GLK targets in five species, found that (perhaps surprisingly?) there was a large divergence in targets, and then wrote a paper from the position of cistrome reprogramming during plant evolution to account for it. But how many of these non-conserved targets are actually real, or biologically meaningful? The authors use RNA-seq to investigate this, and show that most of the species-specific targets revealed by ChIP-seq are largely not differentially expressed in the TF mutants. Only 4% of the non-conserved GLK targets in Arabidopsis are transcriptionally dependent on GLKs. Doesn't this suggest that a large proportion of targets detected by ChIP-seq could be in doubt, or at least not be biologically meaningful? And hence the conclusion that most "targets" are not conserved across species is not surprising after all?

Take also the data shown in Figure 6. Again, this is well presented, but completely unsurprising: duplicated gene targets that are conserved across species also have duplicated TF binding sites in their promoters. Well demonstrated, but it's kind of obvious.

So, I think the authors decided to wrap all this data up in a package and instead argue novelty from the position of transcriptional reprogramming during plant evolution. The point is not what GLK TFs themselves target, but what this tells us about how TF networks evolve overall. And I feel like this is over-stretching the data a bit, even if the data themselves are informative. Somehow, it doesn't come across as the original aim of the work, when the conclusions are drawn from one family of TF.

Incidentally, I would have thought an interesting experiment to identify true conserved vs species-specific targets would be to take, for example, an Arabidopsis GLK and express it in tomato, and then compare ChIP-seq data against tomato GLK in tomato. If the tomato ChIP-seq data are genuine and reproducible, and a feature of the cistrome rather than the TF itself, then you should also see the same targets irrespective of the specific TF used. The authors could also use additional methods to validate some non-conserved, species-specific targets.

Major comments:

Besides my ambivalence above, I have one significant concern about this manuscript, which is technical in nature, and I wonder whether it might account for some of the data described.

1. Line 87 (and methods), please provide detail on what is meant by "two biological replicates". Are the data derived from two independent transformed lines, or two plants derived from the same transgenic event? If the latter, then of course you expect high correlation between replicates (0.98), but this is not true biological replication. How did you characterise the transgenic lines in terms of transgene expression level before undertaking ChIP-seq? How do you know if you are comparing like with like between species? Because if you are not in some way standardising for expression level (which I accept is hard), then you may well get "extra" low-affinity targets in ChIP-seq that will be represented in some species, but not in others. Likewise, if your replicates within a species are technical replicates of the same transgenic event, then you can't identify spurious targets from genuine ones. Which might explain in part the low conservation of targets between species. The ChIP-seq data are validated (Fig 1) using known photosynthetic targets, but presumably these are the high-affinity, highly conserved core targets that will certainly be discovered. It doesn't demonstrate that all the other "non-conserved" targets are genuine, and not experimental noise.

Minor comments:

1. I think in places the writing is a little sparse - I do appreciate a concise story, but I was left wanting for a little more explanation in places (see below).

2. Line 60, the same is true for rice double mutants: doi.org/10.1007/s00425-012-1754-3

3. Line 62, there is also evidence of subfunctionalisation in rice, see [10.1007/s00425-012-1754-3](https://doi.org/10.1007/s00425-012-1754-3). Also, the founding member of GLK family, GOLDEN2, is expressed in bundle sheath cells of maize and may contribute to C4 development.

4. The introduction closes rather abruptly, but I guess that's just a matter of style.

5. What are the first four columns of Fig 2c meant to represent? G1_leaf, G2_leaf, G1_fruit and G2_fruit are not epigenome features, so the legend does not help my understanding. I assume it is to show the transcriptional start sites but what data is actually being mapped in these panels?

6. Lines 224-232. I do not quite follow this paragraph. Comparing expression levels of conserved and non-conserved GLK targets, how does this tell us anything about "biological functions unrelated to GLK"? The opening sentence itself is not very clear. In lines 227-228, "the average FPKM values of the conserved tomato GLK targets were higher than the species-specific ones" - this may be a bias resulting from the fact that photosynthetic transcripts are very highly expressed, so I am not sure how meaningful it is.

"... and they are not significantly different in the GLK loss-of-function mutant" - so this means that once the photosynthetic targets are removed (the "bona fide" targets, if you like), then there's no difference in mean expression level between WT and mutant. Is this the point you are trying to make? And this may be true on average (Fig 5d) but clearly there are examples of conserved, non-photosynthetic targets (grey triangles) that are differentially expressed (Fig 5c). Line 232, "which is consistent with our hypothesis" - which hypothesis specifically? I think overall this paragraph needs more thorough explanation and reasoning, because the more I read it, the more confused I become.

7. Given the wealth of putative target data the authors have uncovered, is it possible to predict GLK binding motif(s)?

Reviewer #3:

Remarks to the Author:

The manuscript, "Cross-species comparison of transcription factor binding reveals wide-spread neutral

cistrome reprogramming during plant evolution” by Shen et al, aims to elucidate the conservation and/or evolution of the targets of two transcription factors (Golden2-like (GLK) TFs) known to be required for chloroplast formation and development, a crucial and highly conserved organelle in plants. The authors examined GLK targets in 5 representative plant species and in one case, tomato, also examined targets in two organs, leaves and fruit. The manuscript describes that the ChIPseq data was robustly validated by three Quality Control evaluations and checked by examining previously known targets. They identified ca 350- over 900 conserved genes targeted by both of the GLKs in the five species. The authors report significant and conserved enrichment of gene targets in the categories of photosynthesis and chlorophyll biosynthesis but also report richly diverse targets in genes and regulators with many other functions and many if not most of these are not conserved among the 5 species examined. This observation allowed the authors to decipher how targets diverged or were conserved over different scales of evolutionary time and processes and led them to the conclusion that the targets having to do with photosynthesis were the most conserved. The manuscript provides confirmation that GLK binding to conserved genes is consequential in that glk double mutants have significantly reduced expression of these genes; expression of GLKs therefore is highly important for a set of conserved functions and results with a pair of targets in Figure 6 shows that GLK binding is substantially preserved during genome evolution and rearrangements in tomatoes.

The manuscript makes a convincing argument based on relatively sound data that GLK expression and preservation of GLK targets in key genes associated with a vital function in plants, photosynthesis, are preserved across species and over evolution and selection. The manuscript specifically addresses an important but somewhat understudied aspect of gene regulation, the conservation of transcription factor targets that are impactful for gene expression. The manuscript includes foundational observations and it will be interesting to see if some of the nuances of GLK regulation, including any specialized functions of either GLK1 and GLK2 can enhance our understanding of how chloroplast and photosynthesis are specifically regulated in particular tissues or parts of plants. The Discussion section of the manuscript nicely predicts analysis that could come from further exploration of the information provided in the manuscript.

Specific comments about information in the manuscript:

Lines 73-77 describe the constructs of GLKs used to evaluate targets in immature tomato fruit. The authors rightly note that overexpression of SIGLK2 causes co-suppression of the endogenous SLGLK2 and so they use the Micro-tom variety for expression of the SIGLK2-GFP construction since Micro-tom has a mutation in its SIGLK2. However, how do they eliminate the possibility that co-suppression alters the availability of cis regulatory sequences by ChIPseq for detection by the SIGLK2-GFP transcripts since Micro-tom continues to produce a (nonfunctiona) SIGLK2 transcript?

Lines 326-327 say that the tomato glk1/glk2 double mutants were provided but do not say what was the background that this mutant line is in and what the mutation specifically is for each of the GLKs. Figure 2 could be improved with more explanation of the correlation comparison in 2b – what subdivisions of data being compared on the two dimensions? Also in figure 2c, what are the “genes” on the vertical axis of the plots (are they the leaf and fruit tomato genes or genes from all the ChIP seq analysis?)?

Figure 3 is an very good depiction of targets of GLKs uncovered by the analysis. However, it was not clear whether the tomato data included results from only tomato leaves or also from tomato fruit. This probably should be explicitly stated.

I assume the data in Figure 5 includes only the data from leaves of tomato, not fruit.

The analysis of sequences in the syteny blocks containing the LHC genes on tomato chromosome 3 and 6 described in Figure 6 is convincing support for the hypothesis that GLK binding is under strong selection pressure for preservation in key conserved GLK target genes.

One of the interesting nuances of the targets of the two GLKs is that there are some tissues where there could be specialized targets of GLK2; in C4 tissues of maize and in developing tomato fruit and Arabidopsis siliques only 1 of the GLKs, GLK2, is expressed. While it is true that in fruit/siliques expression of either GLK1 and GLK2 seem to be capable of targeting similar genes, does the analysis in this manuscript lead the authors to suggest possible explanations for any specialized functions of GLK2 (or GLK1) in specific contexts or organs? Does the data in figure 4 and 5b about maize showing far less conservation of photosynthesis-related targets in maize relate to possible differences between

GLK1 and GLK2? Also does the double mutant analysis in Figure 5 shed light on specialized GLK functions?

Reviewer #4:

Remarks to the Author:

This is an interesting and timely study to investigate conservation of TF binding targets across the plant kingdom and provide insight into what may constitute a 'core' cistrome for a TF. The combination of multi-species TF ChIP-seq and mutant analysis in this study indicates that at least for this particular TF a small subset of the target genes are conserved across monocots/dicots and are highly enriched for GLK's primary role in regulation of chloroplast development and photosynthesis. The duplicated GLK target gene analysis is also quite interesting as it shows how duplicated genes frequently retain the same GLK regulation even over long evolutionary periods. The manuscript is well-written and clear, the analysis and statistical methods are appropriate, and I feel it would be of general interest to Nature Communications readership.

The two major points that I think should be addressed are:

1. Cross-species TF motif analysis: The authors' do not present motifs derived from the ChIP-seq data from the different species. I think they should include this information at a minimum. I think there is also an opportunity to do a bit more motif-binding site analysis to answer an important unaddressed question regarding how the different plants show so many distinct TF target genes. Specifically, the differences in TF target gene profiles could be due to a) differences in the TF motifs leading to different genes targets or b) the motif has not changed significantly and the differential target gene profiles are instead due primarily to new binding sites present at these genes. This would shed light on if changes in TF-DNA binding properties (motifs), or redistribution of TF binding sites is a primary driver of the cistrome divergence.

2. In the abstract the author states that "Their cistromes contain a small number of core ancestral regulatory interactions, and a large number of neutral and dynamic ones that serve as building blocks of new regulatory circuits." I don't feel that the analysis was sufficient to support the second half of the statement regarding 'large number of neutral and dynamic ones ...building blocks of new regulatory circuits.' This could be true, and they do point to an interesting example from literature (EIN2 in fruit development), however no examples are shown of this in the GLK data. Also, I don't feel Fig 5b, and it is the only section of the result that directly refers to this question, provides strong support for this conclusion either as binding strength alone can't be used to classify functional binding site. If they were to look in the partially conserved sets (eg dicot conserved set) to identify a few examples of potential 'new regulatory circuits' this could address this. Otherwise, I think this statement should be removed or modified in the abstract as it implies this is a major finding supported by the analysis which I don't feel it is in its current form. Similarly, the title includes the statement "wide-spread neutral cistrome reprogramming". The use of the word 'neutral' implies to me that most of the non-deeply conserved sites are non-functional which they haven't demonstrated. As the manuscript focuses primarily on using the deeply-conserved TF target genes to identify core transcriptionally and biologically relevant target genes I feel a title change that highlights conservation rather than divergence would be more appropriate.

Reviewer #1 (Remarks to the Author):

This manuscript deals with a comparison of GLK transcription factor (TF) targets in different flowering plant species. The topic is very interesting : understanding the evolution of TF targets is fascinating and timely topic and reports are still rare (doi: 10.1093/molbev/msv210 should be mentioned). It focuses on the GLK TF that has a series of well established targets in photosynthesis genes as well as additional newly evolved specialized functions.

Overall, the message of the paper is convincingly supported and well illustrated. The GLK TFs share only a subset of conserved targets and they are mostly present within conserved binding. The binding conservation not associated with regulation is quite intriguing and could be commented and examined closer. Some of the conclusions of this work might appear almost trivial as they are logic but I truly believe that such quantitative study where the targets have been examined experimentally combining binding (ChIP) and regulation (RNA-seq in mutants) is essential so that rules can be established and not just anticipated. To me, this is the main significance of this work that otherwise does not really involve any scientific or technological breakthrough.

Here are a series of comments

Major points:

1. I am surprised that GLK binding sites models have not been to identify whether the binding could be predicted without performing ChIP or to test whether conserved and less conserved targets have different types of binding sites

Thanks for the suggestion. We have added motif enrichment analysis for all GLK binding sites (Figure 2A), as well as enrichment analysis for the conserved and non-conserved sites (Sup Fig 2). We found that they all have the RGATTTY motif enriched, which is the same as the GLK binding motif reported previously (PMID: 19376934 & 30002259).

Figure 2a:

Sup Fig 2:

We also divided the GLK target genes into five groups. Group 5 contain genes that are conserved in five species, group 4 contain genes that are conserved in 4 species etc. We then checked how many genes in each group would contain the RGATTTY motif hit in their promoter (Sup Fig 5). We also extracted all motif hits and compared their motif hit score distribution. The result showed that the conserved and non-conserved genes have similar GLK TFBS sequences.

Sup Figure 5:

The reviewer also raised an interesting question: whether one can predict TF binding using motif search algorithm without ChIP-seq data. It has been known for both animal and plants

that in silico TF binding prediction would generate too many false positive, since there are many sequences in the genome that match the potential TF binding sites, but are not bound by the TF in vivo. We demonstrated this using the motif hits in the large maize genome. We can found over 2×10^6 sequences that matches the RGATTYY and over 400k GATTCC perfect matches (P5, line 141). This showed that other features such as DNA methylation, chromatin accessibility and DNA conformation could influence TF binding in vivo.

To solve this problem, we trained bag-of-kmer machine learning models to predict GLK binding in all five species. These models outperformed conventional motif search algorithm in binding site prediction. The top K-mers of the models also match the GLK binding motif RGATTYY. The AtGLK1 model is shown in main Figure 2 and below. The remaining models are shown in Sup Fig 1. The model performance scores and their top 10 k-mers are listed in Table S9 and S10, respectively.

It should be noted that the kmer models can not be used to classify conserved and non-conserved targets. Because training needs more features than labels. In the case of GLK, there are only ~100 conserved binding sites (labels) in each species. But there are >1000 k-mers (features). Hence, such model can only be used in predicting the combined TFBS, which have a few thousands inputs for the model to learn. We also tried different deep learning tools and none of them would work on such a small input dataset.

Although we can't use machine learning to separate the conserved and non-conserved binding sites, we have examined the nucleotide diversity (π) of the GLK binding sites, using the Arabidopsis 1001 genome resequencing data (Figure 5). If a site is under strong negative/purify selection, it will have low nucleotide diversity. As a control, we first compared the TFBS in ChIP-seq peaks vs RGATTYY motif hits in non-GLK peak open chromatin regions as background. We found that the GLK TFBS's π is significantly lower (one side KS-test $p=1.3 \times 10^{-15}$) in the ChIP-seq peak than those of the motifs hits in open chromatin (ATAC-seq region without GLK binding). This shows that although there are RGATTYY sequences in open chromatin regions, they are not bound by GLK and not under strong negative selection. We then compared the π in conserved and non-conserved GLK binding sites. The conserved ones also have lower π , confirming that they are under stronger negative selection than the unconserved sites. Finally, we compared the π in DEGs and non-DEGs, and found it is not significantly different ($p=0.59$). This suggest that they experienced similar selection pressure and have functions that were not reflexed in the current gene expression data.

Figure 5:

Finally, we use machine learning to predict the DEGs and nonDEGs using quantitative genomic features such as GLK binding strength, binding sites' distance to gene TSS, gene's expression level in wild-type leaf and which TF have co-binding peaks near the GLK binding sites. It turns out that the TF co-binding information, which we got from the published Arabidopsis cistrome database (DAP-seq data) is crucial for the model accuracy (Figure 7). We also extract the top 10 co-binding TFs that contribute most to the model accuracy, and found that they are some well-known regulators of photosynthesis. This further suggests that individual TF binding might not be sufficient to explain the complex transcription regulation, and we must consider other TFs that co-regulate the target genes, as well as other quantitative features. We have revised our discussion to emphasize this point.

Figure 7:

2. The introduction states that some new roles have been acquired in tomato but then the paper focuses on conserved targets and much less on newly gained targets (bound and regulated). It would be nice to examine whether some true targets specific to one or a few related species lead to some new functions.

Thanks for spotting that. We have now rewritten this part to make it clear. We didn't mean the new role is created by new GLK binding/target. The paper we cited (PMID: 24510723) found that the new role of GLK is the result of tomato SIGLK1 being turned off in fruit tissue, while the remaining SIGLK2 has a latitudinal gradient expression pattern, which then

generated a developmental (ripening) gradient. So the GLK target genes can have a developmental gradient during tomato ripening.

In order to test whether GLK target can acquire new function, one can infer functional divergence from the tissue expression pattern of the duplicated GLK target gene. If the duplicated gene is expressed in a different tissue, it might have acquired a new role there. But the tomato genome is not ideal for tracking GLK targets after duplication, as its genome duplicated ~70 MYA and most of the duplicated GLK targets have been lost. Therefore, we have compared the duplicated maize GLK targets in its subgenome1 and subgenome2, which only separated ~ 10 MYA. We found that losing/gaining of GLK binding in maize genes often lead to alter gene expression pattern, which could suggest that those genes have evolved new biological functions (Figure 8).

Figure 8:

Minor points and suggestions

1. L1 and 32: “neutral” seems inappropriate. I understand that “neutral” is used here because of all the non-conserved targets but I am unsure “non-conserved” equates “neutral”. Among the specific targets, many could have evolved under positive selection but this is not tested (and probably not testable with the data at hand?).

I agree with the reviewer that it is not possible to test for positive selection. We have now added the nucleotide diversity analysis of the AtGLK binding sites using the 1001 Arabidopsis genome data as mentioned above. It showed that the GLK binding sites are under negative selection as they have lower nucleotide diversity. We have stopped using the word “neutral” in the manuscript.

2. In the title “reprogramming” is somewhat obscure. To me it would mean that a given existing function is achieved via a modification in the regulatory network but this is not demonstrated in this study. Perhaps a fairer title could be: “Limited cistrome conservation in cross-species comparison of transcription factor binding implied/suggest wide-spread cistrome reprogramming during plant evolution”

Thanks for the suggestion. We changed the title to: “Limited conservation in cross-species comparison of GLK transcription factor binding suggested wide-spread cistrome divergence”

3. Abstract : L27 what is a primary TF ? L28 dispensable for the expression of the non-conserved ones. L30 : explain the gamma duplication.

My apology for the poor writing. We have deleted those sentences and rewritten the abstract. Previously, we meant that the primary TF (GLK) is dispensable for the expression of some of the non-conserved target genes. The gamma WGD is the genome duplication pre-date monocot–dicot divergence, and it is also referred to as the pan-eudicot palaeohexaploidy WGD.

4. The fact that ChIP-seq peaks come from overexpression constructs of GLK genes may inflate the number of non-conserved targets because the TF overexpression may drive binding where it would not occur in a normal plant. What do the authors think about it? It is a very good point! We tried 2 different ENCODE2 peak calling pipelines and used different q-val cut off (Supplementary table 13). If our conservation analysis is affected by the weak binding sites, raising the cut off would increase the conservation score. But the result is the opposite, that the conservation level decreased when we used a more stringent cut off, suggesting that it is robust.

We have also added one paragraph to the discussion to discuss the limitation of our approach. High protein level can increase the number of binding sites being detected as the weak ones can be called. But a poorly performed ChIP-seq with low enrichment would also reduce the number of TFBS being found, no matter it is using an overexpression line or antibody against the native protein. It is a tricky balance, and the same problem is presented to those researcher that compared the animal and yeast TFs (PMID: 20378774 also reviewed in PMID: 20864205). Hence, it has been suggested that it is better to detect as many sites as possible and then consider the binding quantitatively. Besides that, the weak ones are not always non-function. We have added a few more references about the function of weak TF binding sites. For example, the famous HOX TF paper in Cell 2015 showed that replacing the weak binding site with a high affinity one lead to misexpression.

5. In the abstract “Our study reveals the TF binding dynamics in plant species diverged over 100 million years” I am not sure what is meant here, and how it relates with the results. Could you clarify?

Sorry about that. The new abstract no longer has that sentence.

6. L 165: I do not understand how 646 targets can be conserved between the 5 species and then only 136 of the tomato targets be conserved in all 5 species. Those 646 should be present or did I miss something ? This point needs to be clearer.

It is the sum of conserved target genes in 5 species, we then divided it by the sum of all GLK targets in five species to calculate the average conservation rate. It is indeed a bit confusing. We have now rewritten this sentence and only described the number of conserved genes in each species one by one, instead of adding them up for all five species (page 9, L239)

7. L39 : I would replace “...species with similar genes to create...” by “...individuals within a species to evolve...” because species already have different phenotype.

We changed it to “species or individuals”. Because some unique phenotypes in different species can also be generated by different cis-regulatory elements. Same usage could be

found in PMID: 20864205 and 16000021 when researchers were comparing human and chimpanzee genome.

8. L66 to 92: a lot of this could be in method

Thanks for the suggestion. We have moved them to method.

9. L105: "epigenome features" needs to be defined

We change it to: "...different epigenetic features such as chromatin accessibility, DNA methylation and histone modifications (Fig. 2d). The GLK binding regions display typical active chromatin signatures such as high chromatin accessibility and active histone marks H3K4me3 and H3K27ac usually marking promoter and enhancer regions. They also showed hypo-DNA methylation in CG, CHG and CHH contexts that are often associated with TF binding sites."

10. L204: in figure 5b: enrichment is not define. I assume it is FPKM (?)

We now labelled the y-axes as "GLK ChIP-seq signal". In the figure legend, we said "The y-axis showed the averaged GLK ChIP-seq signal foldchange (log 10) at the peak summit."

11. L227 remove "(Figure 5c and d)". L229: add Figure 5d.

Corrected.

12. L243: "are not located in the previously reported sub-genome regions that resulted from the recent triplication" what is this triplication? Didn't it affect the full genome? a reference may be needed here.

It is whole genome triplication.

Yes, it is a whole-genome event. We changed it to "from the recent genome triplication" and we have now cited the 2012 tomato genome paper there.

13. L279: strong selection -> strong negative selection

14. L291: "has" is missing.

Both corrected.

15. L35:, trimmed with which tool and what are these options?

Changed to "trimmed with the bowtie2 parameter "-3 100"

16. Fig. 5a: X value or X=1 but not X1. The term motif is not clear. It is usually used for TFBS.

Corrected.

Reviewer #2 (Remarks to the Author):

This manuscript by Wei Shen et al describes a comprehensive effort to map the targets of GLK transcription factors in five angiosperm species using ChIP-seq. This is quite an achievement, and a more realistic description of the work than the grand title suggests.

Once you dig down to the results, I think the title might be over-selling the work somewhat, but we all need to grab attention somehow, so I don't have any real problem there.

I have to be honest, I am ambivalent about this manuscript. There's quite a lot of work in it, and it is well presented, but I feel it is written as a bit of a "just-so story". It seems like the

authors set out to compare the GLK targets in five species, found that (perhaps surprisingly?) there was a large divergence in targets, and then wrote a paper from the position of cisome reprogramming during plant evolution to account for it. But how many of these non-conserved targets are actually **real**, or **biologically meaningful**? The authors use RNA-seq to investigate this, and show that most of the species-specific targets revealed by ChIP-seq are largely not differentially expressed in the TF mutants. Only 4% of the non-conserved GLK targets in Arabidopsis are **transcriptionally dependent** on GLKs. Doesn't this suggest that a large proportion of targets detected by ChIP-seq could be in doubt, or at least not be biologically meaningful? And hence the conclusion that most "targets" are not conserved across species is not surprising after all?

I totally agree with the reviewer that our data indeed showed that a large proportion of TF targets are not "transcriptionally dependent" on the TF. This phenomenon is actually well known in yeast and animals, as early as 2010 when ChIP-seq was first used to compare TF in different animal species (Schmidt et al. 2010 Science PMID: 20378774, also reviewed in Dowell 2010 Trends in Genetics PMID: 20864205). We now cited another review by Spivakov (PMID: 24888900), which focus on this issues and its title is "Spurious transcription factor binding: non-functional or genetically redundant?". This phenomenon have also been confirmed by the large-scale TF ChIP-seq studies conducted by the ENCODE and modENCODE. For plant TFs, these is also a nice review paper just came out recently in Cur Opin Plant Biol (PMID: 35679803). We have now cited those paper in the introduction and discussion.

Page 2 L54 to P3 L76:

These studies have shown that homologous TFs with conserved biological function shared few binding sites in different species. For example, a pioneer study in yeast found that a pseudohyphal development-related TF binds only ~20% of the same target genes in comparison among three *Saccharomyces sensu stricto* species⁷. ChIP-seq of two conserved hepatic bZIP and HB TFs in the liver tissues of five vertebrates (human, mouse, dog, opossum and chicken) also found that less than 10% of the binding sites are conserved⁸. The recent large-scale TF analysis conducted by the ENCODE projects also reported a low degree of conservation of TF footprints between human and mouse^{9,10}, suggesting that the animal cisome is highly dynamic during evolution.

In the unicellular eukaryote model organism yeast, exhaustive ChIP-chip experiments have identified the binding sites of all known TFs¹¹, and their impacts on transcription were measured using TF knockout strains^{12,13}. Surprisingly, it was found that many TF binding sites have no transcriptional effect under given conditions. Further TF studies in higher eukaryotes with larger genomes have confirmed that TFs can bind to an unexpectedly large number of sites and most of them has little impact on transcription, suggesting high redundancy and system robustness of the transcription regulatory network^{9,14-18}. Interestingly, those conserved TF binding sites identified in the cross-species comparisons often have the strongest impacts on nearby gene expression^{8,19}. In addition, TF binding sites with strong regulatory potential are often located in super enhancer regions, which are genome hotspots targeted by multiple co-binding TFs¹⁵. Therefore, it has been suggested that multiple TFs jointly contribute to the transcription output in a quantitative manner. Individual TF binding sites may be insufficient to explain transcription, and a cluster of binding sites in the enhancer is key to achieve a precise and robust transcription regulation²⁰.

For the terms "real", "biologically meaningful" and "transcriptionally dependent" suggested by the reviewer, I believe the last one is more accurate and we should avoid using the first two. As suggested in the above reviews in yeast and animal field, ChIP-seq can not be used to find "real" or "biologically meaningful" binding sites. It can only find reproducible binding sites (enriched against a background of noise within a false discovery rate). They also emphasized that ChIP-seq is an enrichment assay. Its data often contain more false negative than false positive (e.g. if the enrichment is low, the weak sites will not be called).

Also, many large-scale plant TF ChIP-seq studies have found ~10k binding sites for each TF (PMID: 27811239, PMID: 27203113). Same for the animal TFs data from ENCODE and modENCODE. It will be unimaginable that all those published ChIP-seq targets are “not real” simply because most of them are not transcriptionally dependent of the TF.

In addition, those animal studies mentioned above, as well as a recent review paper for plant TF (PMID: 35679803), have shown that RNA-seq itself can be misleading. The non-DEG might be DEG in a different condition, different cell type or different growth stage. Also, if two or more TFs can bind to the promoter of a gene, losing one TF will not affect its expression. Therefore, those “genetically redundant” TF binding would be “real” and “meaningful” under some conditions.

Also, our cross-species analysis found not all PS genes are conserved, and some conserved PS genes could be nonDEGs. For example, chloroplast ATP synthase subunit ATPC1 is a conserved GLK target with strong binding sites (Sup Fig 6). It is not a DEG in the Arabidopsis mutant, but it is a DEG in the tomato mutant. Therefore, it is better to use the last term “transcriptionally dependent” or “regulatory potential” to describe TF binding sites.

Take also the data shown in Figure 6. Again, this is well presented, but completely unsurprising: duplicated gene targets that are conserved across species also have duplicated TF binding sites in their promoters. Well demonstrated, but it's kind of obvious.

Thanks for the suggestion, we have removed panel c and d from the figure and rewritten the manuscript text. Fig 6a-b showed that the conserved tomato GLK targets are duplicated > 100 MYA instead of 60 MYA. Because for tomato, we can calculate the Ks of the synteny blocks to infer when the duplicate occurred. Inspired by the reviewer’s comment, we also added a new data from maize duplicated GLK targets into this figure. Unlike the tomato genome, maize has the a recent duplication and their genes can still be split into subgenome 1 and subgenome 2 (PMID: 21368132). We found that those duplicated GLK target genes have already altered their GLK binding in such a short time span ~5-10 MYA. In addition, those duplicated genes that lost GLK binding have evolved a different tissue expression pattern as indicated by the low expression correlation (below).

So, I think the authors decided to wrap all this data up in a package and instead argue novelty from the position of transcriptional reprogramming during plant evolution. The point is not what GLK TFs themselves target, but what this tells us about how TF networks evolve overall. And I feel like this is over-stretching the data a bit, even if the data themselves are informative. Somehow, it doesn't come across as the original aim of the work, when the conclusions are drawn from one family of TF.

I agree with the reviewer. We have re-written our discussion about the TF network evolution part and remove the old figure 7 from the revised manuscript. The reviewer is right that we didn't start this research to understand GLK function, as we are not a research group working on photosynthesis or plant biology. Instead, we work on genome and functional genomics data, and we want to understand how plant genes are regulated and how their cistrome evolved. Hence we pick a well-known TF (GLK) for the cross-species comparison.

We also agree with the reviewer that our previously proposed cistrome evolution model is only supported by the GLK data, and other TF might not act like this. This actually highlights why more of this comparative TF study is needed, as the field needs more TF binding data from various species to get a better picture of plant cistrome evolution.

Also, reviewer#1 reminded us that there is one paper comparing MADS-box TF binding in *A. thaliana* and *A. lyrata*, the result of which is consistent with our data. They found ~20% TF target conservation in 2 closely related species separated only ~6 MYA. But they did not have mutant RNA-seq data to check the conserved binding site's impact on gene expression. If they have, I believe they will find something similar to what we saw in the case of GLK. We have now added this to the discussion.

In addition, we added 2 new results to support our case that the observed GLK binding divergence is real, despite the fact that many binding sites are near non-DEGs:

- 1) Nucleotide diversity analysis showed that GLK binding sites in DEG and nonDEG are under similar negative selection pressure.
- 2) Machine learning to predict the transcriptional effect of GLK binding site using quantitative genome features (ChIP-seq signal fold change, binding distance to gene, gene's initial expression level), plus co-binding TFs. It showed that TF co-binding is an important factor for the model accuracy.

Incidentally, I would have thought an interesting experiment to identify true conserved vs species-specific targets would be to take, for example, an *Arabidopsis* GLK and express it in tomato, and then compare ChIP-seq data against tomato GLK in tomato. If the tomato ChIP-seq data are genuine and reproducible, and a feature of the cistrome rather than the TF itself, then you should also see the same targets irrespective of the specific TF used. The authors could also use additional methods to validate some non-conserved, species-specific targets.

Thanks, I think that is a very good experiment. We have now expressed the maize ZmGLK1-HA in *Arabidopsis* and performed ChIP-seq, and it turns out that ZmGLK1 binds exactly like the *Arabidopsis* GLKs (Figure 6). So it is the cistrome variation that determines how TF binds. For example, *Arabidopsis* GLK1 can bind to the chlorophyll biosynthesis gene HEMA and

we can find the GLK binding motif in its promoter open chromatin. The maize HEMA gene is not targeted by GLK. When we expressed the maize GLK in Arabidopsis, we can find ChIP-seq signal in the Arabidopsis HEMA promoter. We also group the At genes into conserved GLK target (left column), At only (middle column) and Zm only (right column). The heat map showed the ZmGLK1-HA in Arabidopsis ChIP-seq signal in these 3 groups, and the maize only group (right column) has no binding, confirming that the cis-regulation element in the genome determined the TF binding.

Figure 6:

As to the question whether we can validate our ChIP-seq using additional method, we have added 4 lines of evidences:

- 1) Motif enrichment analysis showed that both the conserved and non-conserved TF targets have the GLK binding motif (Sup Fig 3 to 5). Same for the DEG vs nonDEG, and strong vs weak TF binding sites. If those bindings are not real, they should not have the GLK motif enriched.
- 2) Nucleotide conservation analysis. If the binding is “real and functional”, they might be under stronger negative selection than those “unreal” binding. So we used the 1001 Arabidopsis genome resequencing data and calculate the nucleotide diversity of GLK binding sites in all chip-seq peaks and compare that to the background, which is unbound motif hits in the open chromatin regions (Fig 5). We found the sites in chip-seq peaks have significantly lower sequence variations ($p=1.33e-15$). But the sites in DEG and non-DEG are not significantly different ($p=0.597$). This suggest that the nonDEG binding sites could also have important function.
- 3) If some less conserved or non-DEG is “not real” targets due to the weak ChIP-seq signal or just false positive of experiment, we would expect that the “real” strong binding sites will remain strong in the recently duplicated genes. This means that the TF binding strength of the duplicated genes will be highly correlated. We performed Kendall’s correlation test to compare the ranking of GLK binding strength in the duplicated maize GLK targets in subgenome 1 and 2 (page 15, L412). The Kendall’s tau is 0.174 ($p=0.06$), suggesting that the correlation is weak and they are not significantly correlated. We also showed that some well-known PS genes can have strong and weak binding sites in different species. For example, Sup Figure 6 showed that PSII subunit gene *PsbTn* has

strong binding in Arabidopsis (rank 102th), and very weak binding in tomato (rank 1083th). But it is differentially expressed in both tomato and Arabidopsis.

- 4) Finally, we can use a random forest model to predict DEG and nonDEG status using quantitative genome features (binding strength, distance, gene expression) as well as TF co-binding data. This showed that it is best to consider TF binding quantitatively, instead of just “bound” or “unbound”. In addition, it showed that the transcription potential of TFBS or whether the site is genetically redundant, could be due to the co-binding TFs that jointly regulate the target genes.

Major comments:

Besides my ambivalence above, I have one significant concern about this manuscript, which is technical in nature, and I wonder whether it might account for some of the data described.

1. Line 87 (and methods), please provide detail on what is meant by "two biological replicates". Are the data derived from two independent transformed lines, or two plants derived from the same transgenic event? If the latter, then of course you expect high correlation between replicates (0.98), but this is not true biological replication. How did you characterise the transgenic lines in terms of transgene expression level before undertaking ChIP-seq? How do you know if you are comparing like with like between species? Because if you are not in some way standardising for expression level (which I accept is hard), then you may well get "extra" low-affinity targets in ChIP-seq that will be represented in some species, but not in others. Likewise, if your replicates within a species are technical replicates of the same transgenic event, then you can't identify spurious targets from genuine ones. Which might explain in part the low conservation of targets between species. The ChIP-seq data are validated (Fig 1) using known photosynthetic targets, but presumably these are the high-affinity, highly conserved core targets that will certainly be discovered. It doesn't demonstrate that all the other "non-conserved" targets are genuine, and not experimental noise.

Thanks for raising this point. It is indeed worth discussing. First, it has been shown that we should not consider that those weak binding is “non-functional” or not genuine (PMID: 24888900). To further emphasize this, we added another discussion paragraph describing a famous study of the drosophila HOX binding site published in Cell 2015. They found that weak binding is key to achieve specificity in transcription regulation. If the weak HOX is replaced by a high affinity site, the gene expressed will be disrupted. In addition, we also cited a recent nature genetic paper about 4 stem cell TF binding sites, in which they found that the spacing between those sites are also important.

We have already used a very stringent ENCODE2 pipeline (MACS2-IDR <0.01) and the peaks we did not use (IDR >0.01 and <0.05) still have GLK motif enrichment (Sup Fig 3). Many published plant TF studies used much less stringent cut off and they would easily call >10k target genes for each TF (e.g. PMID: 27811239, PMID: 27203113). It is unlikely that all of their target genes will be differentially expressed when the TF is knockout. In addition, as the sensitivity of ChIP-seq improves, more “weak” TFBS could be found. Hence, it is better to consider TF binding quantitatively, instead of bind and not bind.

I am afraid that the correlation in the ENCODE2 QC pipeline should not be used like this. Because the deepTool multibamssummary software would first divide the genome into 10kb bins, and count the chip-seq reads in each bin. With a few gene rich bins with high counts and lots of bins with zero counts, their pearson correlation should be very high unless there is something wrong with the data. Hence it can only be used for ChIP-seq QC. If we want to compare how similar are the 2 TF binding profiles, we should use the union of the peaks called from 2 CHIP-seq, then calculate the read counts in each peaks and perform a correlation test. For example, we have done such comparison in page 6 line 180. The correlation of tomato GLKs is only 0.89. Alternatively, some tools would prefer to use gene promoter region or the ATAC-seq region to perform the comparison. Each has their advantage and limits. Also, independent ChIP-seq performed on a plant/transgenic line with the same genetic content is considered as biological replicates (e.g. FEA4 PMID: 25616871 and KNOOT1 PMID: 19567707). The ENCODE has a stricter terminology and called it "isogenic replicate" but this word has not been used in plant. As discuss above, I agree with the reviewer that no one can control for protein level within / between experiments, which is a major limit of such cross-species comparison. We have now added a discussion paragraph about the limit of this study. Although we can not control for protein level, ChIP-seq data generated using overexpressed TF is far more accurate than binding site predicted/detected by in silico sequence analysis, yeast one hybrid or DAP-seq. For example, a recent paper in nature plant also used the same overexpressed TF line for ChIP-seq to compared WRKY TF in liverworts and arabidopsis (PMID: 34045707), and 2 well-known nitrogen metabolism TF regulatory networks published in nature and nature communications were constructed using only Y1H and gene expression data (PMID: 30356219 & 30952851). The same problem was also encounter by those yeast and animal TF comparison studies (Schmidt et al. 2010 Science PMID: 20378774, reviewed in PMID: 20864205). Same for ENCODE project's human and mouse TF comparison. Hence, the field generally accept that ChIP-seq enrichment will greatly affect how many weak sites will be detected. As a result, most TF ChIP-seq studies now days would try to identify as many TF binding sites as possible, and consider their binding strength quantitatively.

Minor comments:

1. I think in places the writing is a little sparse - I do appreciate a concise story, but I was left wanting for a little more explanation in places (see below).

Thanks for the suggesting, we have rewritten the introduction and discussion as suggested.

2. Line 60, the same is true for rice double mutants: doi.org/10.1007/s00425-012-1754-3

Thanks! We have now included this into our introduction.

3. Line 62, there is also evidence of subfunctionalisation in rice, see 10.1007/s00425-012-1754-3. Also, the founding member of GLK family, GOLDEN2, is expressed in bundle sheath cells of maize and may contribute to C4 development.

Thanks again. We have added both the rice and maize paper to the introduction.

4. The introduction closes rather abruptly, but I guess that's just a matter of style.

Thanks for the suggestion. We have add a summary paragraph to the end of the introduction.

5. What are the first four columns of Fig 2c meant to represent? G1_leaf, G2_leaf, G1_fruit and G2_fruit are not epigenome features, so the legend does not help my understanding. I assume it is to show the transcriptional start sites but what data is actually being mapped in these panels?

Sorry for the poor labelling. The first 4 columns are the GLK1/2 ChIP-seq signals in leaf and fruit tissues. They are not epigenome features, but we wanted to show the epigenome features near them, and all the heatmap data is clustered using the data in the first column. We now stated in the figure legend: "Heatmap and average signal plot showing the tomato epigenome features near the GLK binding sites. From left to right, GLK1 ChIP-seq in leaf, GLK2 ChIP-seq in leaf, GLK1 ChIP-seq in fruit, GLK2 ChIP-seq in fruit, chromatin accessibility, H3K4me3, H3K27ac, and CG, CHG and CHH methylation. Regions 2 kb up and downstream of the gene transcriptional start sites is shown."

6. Lines 224-232. I do not quite follow this paragraph. Comparing expression levels of conserved and non-conserved GLK targets, how does this tell us anything about "biological functions unrelated to GLK"? The opening sentence itself is not very clear. In lines 227-228, "the average FPKM values of the conserved tomato GLK targets were higher than the species-specific ones" - this may be a bias resulting from the fact that photosynthetic transcripts are very highly expressed, so I am not sure how meaningful it is.

"... and they are not significantly different in the GLK loss-of-function mutant" - so this means that once the photosynthetic targets are removed (the "bona fide" targets, if you like), then there's no difference in mean expression level between WT and mutant. Is this the point you are trying to make? And this may be true on average (Fig 5d) but clearly there are examples of conserved, non-photosynthetic targets (grey triangles) that are differentially expressed (Fig 5c). Line 232, "which is consistent with our hypothesis" - which hypothesis specifically? I think overall this paragraph needs more thorough explanation and reasoning, because the more I read it, the more confused I become.

Thank you for spotting this. We have now removed this paragraph, and refocused that result section to the new maize subgenome comparison.

7. Given the wealth of putative target data the authors have uncovered, is it possible to predict GLK binding motif(s)?

Yes, we have added motif enrichment analysis as well as k-mer machine learning models to predict GLK binding motif/kmers.

Reviewer #3 (Remarks to the Author):

The manuscript, "Cross-species comparison of transcription factor binding reveals widespread neutral cistrome reprogramming during plant evolution" by Shen et al, aims to elucidate the conservation and/or evolution of the targets of two transcription factors (Golden2-like (GLK) TFs) known to be required for chloroplast formation and development, a crucial and highly conserved organelle in plants. The authors examined GLK targets in 5 representative plant species and in one case, tomato, also examined targets in two organs, leaves and fruit. The manuscript describes that the ChIPseq data was robustly validated by three Quality Control evaluations and checked by examining previously known targets. They identified ca 350- over 900 conserved genes targeted by both of the GLKs in the five species. The

authors report significant and conserved enrichment of gene targets in the categories of photosynthesis and chlorophyll biosynthesis but also report richly diverse targets in genes and regulators with many other functions and many if not most of these are not conserved among the 5 species examined. This observation allowed the authors to decipher how targets diverged or were conserved over different scales of evolutionary time and processes and led them to the conclusion that the targets having to do with photosynthesis were the most conserved. The manuscript provides confirmation that GLK binding to conserved genes is consequential in that glk double mutants have significantly reduced expression of these genes; overexpression of GLKs therefore is highly important for a set of conserved functions and results with a pair of targets in Figure 6 shows that GLK binding is substantially preserved during genome evolution and rearrangements in tomatoes. The manuscript makes a convincing argument based on relatively sound data that GLK expression and preservation of GLK targets in key genes associated with a vital function in plants, photosynthesis, are preserved across species and over evolution and selection. The manuscript specifically addresses an important but somewhat understudied aspect of gene regulation, the conservation of transcription factor targets that are impactful for gene expression. The manuscript includes foundational observations and it will be interesting to see if some of the nuances of GLK regulation, including any specialized functions of either GLK1 and GLK2 can enhance our understanding of how chloroplast and photosynthesis are specifically regulated in particular tissues or parts of plants. The Discussion section of the manuscript nicely predicts analysis that could come from further exploration of the information provided in the manuscript.

Specific comments about information in the manuscript:

Lines 73-77 describe the constructs of GLKs used to evaluate targets in immature tomato fruit. The authors rightly note that overexpression of SIGLK2 causes co-suppression of the endogenous SLGLK2 and so they use the Micro-tom variety for expression of the SIGLK2-GFP construction since Micro-tom has a mutation in its SIGLK2. However, how do they eliminate the possibility that co-suppression alters the availability of cis regulatory sequences by ChIPSeq for detection by the SIGLK2-GFP transcripts since Micro-tom continues to produce a (nonfunctional) SIGLK2 transcript?

That is indeed a good point, given the fact that multiple tomato loss-of-function mutants turn out to be negative dominant and gain-of-function, we should be more careful. The glk2 mutation in microTom caused a frame shift in the first coding exon. We can not be 100% sure that truncated protein would not compete with the transgene. Since it no longer has the conserved DNA binding domain, we can only say that it is unlikely that the truncated protein can bind to DNA. We also compared our tomato GLK1 and the GLK2 data (page 6, line 171-184), and found that the GLK2 data has more peaks called by MACS2-IDR. We then use the union of GLK1 and GLK2 peaks and calculate their read counts, and found they are highly correlated (pearson correlation coefficient = 0.89). This suggests they do have the same binding sites, and the GLK2 chip-seq experiments have better signal to noise ratio (hence more weak binding sites being called). Therefore, it would suggest that the truncated endogenous SIGLK2 in MT can't compete with the over-expressed GLK2-GFP for those binding sites. Otherwise, we might get more peaks in the GLK1 data.

Lines 326-327 say that the tomato glk1/glk2 double mutants were provided but do not say what was the background that this mutant line is in and what the mutation specifically is for each of the GLKs.

We now added it to the material section that the tomato double mutant we got from the southwest university stock centre is in the microTom background. We also sequenced its crisper cut sites to confirm the deletion (Supplementary Figure 7).

Figure 2 could be improved with more explanation of the correlation comparison in 2b – what subdivisions of data being compared on the two dimensions? Also in figure 2c, what are the “genes” on the vertical axis of the plots (are they the leaf and fruit tomato genes or genes from all the ChIP seq analysis?)?

My apology for the poorly figure 2 legend. We have revised both the figure and legend. For the heatmap, they are indeed tomato GLK target genes. So we change the axis label to “tomato GLK target genes”.

Figure 3 is an very good depiction of targets of GLKs uncovered by the analysis. However, it was not clear whether the tomato data included results from only tomato leaves or also from tomato fruit. This probably should be explicitly stated.

Thanks for spotting that. Only tomato leaf GLK target genes are used for the 5 species comparison. We have now added it into the figure 3 legend.

I assume the data in Figure 5 includes only the data from leaves of tomato, not fruit.

Yes, we have now added the description into the legend (the original figure for tomato gene has been moved to supplementary data as sup fig. 5, only the Arabidopsis genes are shown in the main text fig. 5).

The analysis of sequences in the syntenic blocks containing the LHC genes on tomato chromosome 3 and 6 described in Figure 6 is convincing support for the hypothesis that GLK binding is under strong selection pressure for preservation in key conserved GLK target genes. One of the interesting nuances of the targets of the two GLKs is that there are some tissues where there could be specialized targets of GLK2; in C4 tissues of maize and in developing tomato fruit and Arabidopsis siliques only 1 of the GLKs, GLK2, is expressed. While it is true that in fruit/siliques expression of either GLK1 and GLK2 seem to be capable of targeting similar genes, does the analysis in this manuscript lead the authors to suggest possible explanations for any specialized functions of GLK2 (or GLK1) in specific contexts or organs?

Yes, I agree with the reviewer. We added a paragraph in the introduction about those reported sub-functionalization of GLK1 and GLK2: “Sub-functionalization of GLKs in monocots such as rice and maize has been reported²². They also play a role in fruit

development, where GLK1 is switched off, and the GLK2 adapts a latitudinal gradient expression pattern resulting in an uneven coloration of the fruit tissue^{23,24}.” But from the binding site perspective, we didn’t observed any major tissue-specific bindings. We have used the tool diffbind to quantitatively compare the leaf and fruit ChIP-seq data, and very few regions could be called as differentially binding with significant p-val.

Does the data in figure 4 and 5b about maize showing far less conservation of photosynthesis-related targets in maize relate to possible differences between GLK1 and GLK2? Also does the double mutant analysis in Figure 5 shed light on specialized GLK functions?

Sorry for the poor color code in the figure 5. The maize GLK target conservation (both in the PS and chlorophyll biosynthesis pathways) is not low. The light color there is a bit misleading. It is actually the enrichment score (adjusted p-val). We have revised the figure and provided a supplementary table S14. The conservation score is calculated by the dividing the conservation events (observed) and all paired wise comparison performed (expected). For the two maize GLKs, they are just differentially expressed into the mesophyll (has GLK1) and bundle sheath (has GLK2) cells. Their binding preference is not changed, as indicated by the motif enrichment analysis. However, the chromatin accessibility differences in mesophyll and bundle sheath cells could potentially alter their binding in a cell-specific manner (PMID: 35758633). One interesting observation is that both tomato and maize used H3K27me3 to repressed GLK1 when they want to switch it off in the fruit and bundle sheath cells.

Table S14: GLK target conservation score

	All GLK target genes	Photosynthesis genes only	Tetrapyrrole biosynthesis genes only
Arabidopsis	0.475625	0.9538462	0.8545455
Niben	0.53159	0.9835616	0.7444444
Tomato	0.4578538	0.9333333	0.725
Tobacco	0.6108434	0.9555556	0.84
Maize	0.4552801	0.95625	0.88

Reviewer #4 (Remarks to the Author):

This is an interesting and timely study to investigate conservation of TF binding targets across the plant kingdom and provide insight into what may constitute a ‘core’ cistrome for a TF. The combination of multi-species TF ChIP-seq and mutant analysis in this study indicates that at least for this particular TF a small subset of the target genes are conserved across monocots/dicots and are highly enriched for GLK's primary role in regulation of chloroplast development and photosynthesis. The duplicated GLK target gene analysis is also quite interesting as it shows how duplicated genes frequently retain the same GLK regulation even over long evolutionary periods. The manuscript is well-written and clear, the analysis and statistical methods are appropriate, and I feel it would be of general interest to Nature Communications readership.

The two major points that I think should be addressed are:

1. Cross-species TF motif analysis: The authors’ do not present motifs derived from the

ChIP-seq data from the different species. I think they should include this information at a minimum. I think there is also an opportunity to do a bit more motif-binding site analysis to answer an important unaddressed question regarding how the different plants show so many distinct TF target genes. Specifically, the differences in TF target gene profiles could be due to a) differences in the TF motifs leading to different genes targets or b) the motif has not changed significantly and the differential target gene profiles are instead due primarily to new binding sites present at these genes. This would shed light on if changes in TF-DNA binding properties (motifs), or redistribution of TF binding sites is a primary driver of the cistrome divergence.

Thanks for the suggestion. We have now added the motif enrichment analysis and the result is shown in Fig2 A. The motif enrichment result of conserved/non-conserved genes, DEG vs non-DEG and strong vs weak binding sites are shown in the Supplementary Figure S3 and S4. As reported in the animal TF comparisons, the TF motif is often the same in different types of binding sites, which could suggested that the surrounding region might encode information, such as motifs of co-binding TFs, that could influence how they regulate nearby gene expression.

To test the hypothesis a and b, we have now transformed the maize GLK1 to Arabidopsis and performed ChIP-seq. The heterologous expressed maize GLK1 followed the endogenous AtGLKs' binding pattern. Therefore, it is indeed the genome cis-regulatory sequence variations that drive the TF binding divergence. So the hypothesis b is correct.

2. In the abstract the author states that "Their cistromes contain a small number of core ancestral regulatory interactions, and a large number of neutral and dynamic ones that serve as building blocks of new regulatory circuits." I don't feel that the analysis was sufficient to support the second half of the statement regarding "large number of neutral and dynamic ones ...building blocks of new regulatory circuits." This could be true, and they do point to an interesting example from literature (EIN2 in fruit development), however no examples are shown of this in the GLK data. Also, I don't feel Fig 5b, and it is the only section of the result that directly refers to this question, provides strong support for this conclusion either as binding strength alone can't be used to classify functional binding site. If they were to look in the partially conserved sets (eg dicot conserved set) to identify a few examples of potential 'new regulatory circuits' this could address this. Otherwise, I think this statement should be removed or modified in the abstract as it implies this is a major finding supported by the analysis which I don't feel it is in its current form. Similarly, the title includes the statement "wide-spread neutral cistrome reprogramming". The use of the word 'neutral' implies to me that most of the non-deeply conserved sites are non-functional which they haven't demonstrated. As the manuscript focuses primarily on using the deeply-conserved TF target genes to identify core transcriptionally and biologically relevant target genes I feel a title change that highlights conservation rather than divergence would be more appropriate.

I agree with the reviewer that this part of the abstract as well as some discussion are a bit hypothetical and lack experimental evidence. We have deleted them and rewritten the abstract and discussion with a stronger focus on TF binding divergence, potential function of weak binding sites and importance of TF co-binding and the quantitative TF occupancy data. The title is now changed to "Limited conservation in cross-species comparison of GLK transcription factor binding suggested wide-spread cistrome divergence". I also agree that we should not call them "neutral". As we didn't mean to say that they are non-functional.

Instead, our nucleotide diversity analysis showed that those sites in nonDEGs are also under negative selection.

Reviewers' Comments:

Reviewer #1:

Remarks to the Author:

This revised manuscript addresses the prediction and comparison of GLK bound regions and regulated genes in different species. It goes well beyond answering the comments of the reviewers of the original manuscript by adding a substantial amount of novel analyses. These new analyses clearly add to the value of this work. The use of machine learning is particularly appropriate and the results from Fig. 7 are really interesting. The AUROC value attesting the power of the binding models are really impressive even if I have some questions about it (see point 1)

Suggestions for improvement

1. The way the ROC analysis was made to predict GLK binding is not sufficiently explained. Was the model trained on part of the data and tested on another part?(sizes of training vs test sets?) Was a model built from one plant used in another one?
2. A DEG can be determined for sure but it is harder to claim a gene is non-DEG as it could depend of the tissue or cell type. A non DEG from leaves can become a DEG in another tissue. So the conclusion about cis element conservation independently of regulation should probably be moire careful.
3. Line 123: overlap with ATAC peaks and claim of binding to open regions cannot be made on the sole basis of a few examples. It requires some genome wide analysis comparing ATAC signal vs ChIP signal in each ChIP peak.
4. Line 180 : a rigorous comparison of GLK1 and GLK2 binding could have been done on the peak union by plotting, for each peak, the GLK1 vs GLK2 coverages. This is more informative than overlap or the combined information presented in Fig. S2
5. Line 212: the same comment applies when identifying peaks that are specific to some species. They might appear specific because they are really weak/absent or just because they did not pass the detection threshold of peak finders. A plot showing the respective coverage between 2 species would be needed
6. The term target genes used throughout the manuscript should be clarified. It seems to mean : bound gene. But target gene is usually used for direct (= bound) DEG

Reviewer #2:

Remarks to the Author:

The authors have made a genuinely impressive effort to improve the manuscript, with a large body of additional analysis and experimentation. The manuscript is much clearer in terms of writing, and the figures are excellent. I am pleased that the authors took the critique on board and made an extensive effort to produce a manuscript that is more compelling, engaging and informative. Well done.

My previous concerns (and more) have been addressed to my satisfaction, in particular the cross-species complementation experiment and the analysis that reveals the importance of TF co-binindg. I only have very minor comments.

Abstract: although generally the writing has been significantly improved, I think the abstract could be refined a little more for clarity. For example, lines 34-36 could be improved thus (just a suggestion, I appreciate word limits may be a problem): "GLK-dependent gene expression is associated with highly conserved binding sites, but many sites near genes that are not differentially expressed in glk mutants are nevertheless under purifying selection. Accordingly, we trained machine learning models to predict the transcriptional outcome of GLK binding sites using multiple..."

Line 66: what are "given conditions"?

Line 432, lyrata

Reviewer #3:

Remarks to the Author:

I found that the authors adequately and appropriately addressed the concerns I brought up in my initial review. I feel that I cannot comment on the issues raised by the other reviewers, however. I think the work and the conclusions presented in the manuscript are important even given the current limitations of analysis. The authors have recognized most of the limitations of their analysis and the manuscript provides an important road map for future work that will will expand the interpretations on TF function and cis regulatory element roles. Aside from a small typographic error in line 215, I did not find obvious errors. The manuscript has been improved considerably in response to the reviewers' comments.

Reviewer #4:

Remarks to the Author:

The revised manuscript resolves all issues I had in the original review. The new analysis, figures and text add valuable new information. Altogether the manuscript provides a very thorough analysis of GLK cistrome conservation and variation, and the impact on gene expression. In particular, the GLK mutant expression analysis of the two factors is powerful and illustrative.

Response to reviewer comments

Reviewer #1 (Remarks to the Author):

This revised manuscript addresses the prediction and comparison of GLK bound regions and regulated genes in different species. It goes well beyond answering the comments of the reviewers of the original manuscript by adding a substantial amount of novel analyses. These new analyses clearly add to the value of this work. The use of machine learning is particularly appropriate and the results from Fig. 7 are really interesting. The AUROC value attesting the power of the binding models are really impressive even if I have some questions about it (see point 1)

Suggestions for improvement

1. The way the ROC analysis was made to predict GLK binding is not sufficiently explained. Was the model trained on part of the data and tested on another part?(sizes of training vs test sets ?) Was a model built from one plant used in another one ?

We used the k-mer grammar software's default (80/20) setting to randomly split the input into 80% training and 20% test datasets. We now stated this in the method section. In the previous manuscript, we didn't check the k-mer model precision using data from a heterologous species. We have now run this analysis and the result (model precision score) has been added to supplementary table S9 and shown below.

	Arabidopsis data	Tobacco data	Tomato data	Rice data	Maize data
AtGLK1 model	NA	0.98	0.89	0.67	0.69
Nben GLK1 model	0.85	NA	0.73	0.79	0.79
Tomato GLK1 model	0.89	0.86	NA	0.66	0.72
Rice GLK1 model	0.95	0.99	0.93	NA	0.96
Maize GLK1 model	0.94	0.98	0.95	0.95	NA

	Arabidopsis data	Tobacco data	Tomato data	Rice data	Maize data
AtGLK2 model	NA	0.93	0.9	0.69	0.73
Nben GLK2 model	0.95	NA	0.96	0.83	0.92
Tomato GLK2 model	0.92	0.9	NA	0.71	0.79
Rice GLK2 model	0.96	0.98	0.97	NA	0.97
Maize GLK2 model	0.98	0.99	0.98	0.98	NA

2. A DEG can be determined for sure but it is harder to claim a gene is non-DEG as it could depend of the tissue or cell type. A non DEG from leaves can become a DEG in another tissue. So the conclusion about cis element conservation independently of regulation should probably be more careful.

I agree. We added in L309: “..some TF ChIP-seq target genes might be differentially expressed in a certain cell-types or under specific treatment when the TF is knockout.” We also added this to the discussion L448: “In addition, the regulatory potential of a TF binding site could not be fully demonstrated based on individual transcriptome analysis, since a non-differentially expressed TF target gene could become differentially expressed in another growth condition or tissue.”

3. Line 123: overlap with ATAC peaks and claim of binding to open regions cannot be made on the sole basis of a few examples. It requires some genome wide analysis comparing ATAC signal vs ChIP signal in each ChIP peak.

Thanks for pointing it out. We have rewritten this sentence to be precise:“suggesting that GLKs bind to open chromatin regions in their gene promoter”

4. Line 180 : a rigorous comparison of GLK1 and GLK2 binding could have been done on the peak union by plotting, for each peak, the GLK1 vs GLK2 coverages. This is more informative than overlap or the combined information presented in Fig. S2

That is a good idea. We now added a Sup Fig. S3 showing the GLK1 and GLK2 ChIP-seq read counts in their peak unions for all 5 species.

5. Line 212: the same comment applies when identifying peaks that are specific to some species. They might appear specific because they are really weak/absent or just because they did not pass the detection threshold of peak finders. A plot showing the respective coverage between 2 species would be needed.

I totally agree with the reviewer that it is indeed a problem of cross-species comparison. The same question has also been raised for the animal and yeast TF studies (PMID 20519030 & 20864205), but there is no other way to call the presence/absence of binding without using a threshold. Besides the plot suggested the reviewer, we made 2 more changes to address this. First, we have revised that paragraph to highlight this issue: “However, it should be noted that ChIP-seq could identify TF binding sites of a wide range of binding strengths. A qualitative analysis of ChIP-seq peaks, which labels a region as bound or unbound, is known to have limitations^{35,36}. For example, weak binding sites are prone to be mischaracterized in a cross-species comparison under different ChIP-seq enrichment and peak caller detection threshold. To compare GLK binding quantitatively between species, we have calculated the ChIP-seq read counts in the promoter of their homologous gene pairs. The conserved GLK target gene pairs have high ChIP-seq signal in both species, while the species-specific ones only have high ChIP-seq signal in one species (Supplementary Fig. 6).”

Second, we added a Sup Fig. 5 showing genome browser tracks of the mentioned species-specific GLK ChIP-seq target genes. Their promoter GLK ChIP-seq signals are significantly different in 5 species.

Lastly, in Sup Fig. S6, we added 5x4 scatter plots showing the GLK chip-seq read coverage (log2CPM) in conserved and species-specific genes for each cross-species comparison as suggested by the reviewer. The yellow dots represent pairs of orthologous genes bound by GLK in two species. The blue dots are pairs of orthologous genes with only one of them bound by GLK. It is clear that the conserved ones have high ChIP-seq signal in both species, while the species-specific ones has low signal in one species.

6. The term target genes used throughout the manuscript should be clarified. It seems to mean : bound gene. But target gene is usually used for direct (= bound) DEG

I agree. The term TF target/bound genes and TF regulated target/genes can be confusing and often mixed used. For example, the first yeast TF network paper in nature called both DEG and non-DEGs as target genes, as long as they have promoter ChIP-seq peak. Same for this 2015 PNAS paper for plant TF titled “Genome-wide identification of CCA1 targets uncovers an expanded clock network in Arabidopsis” (PMID 26261339). Perhaps a more precise way is to call them “ChIP-seq identified target genes” or “ChIP-seq target” for short, and distinguish them from “TF regulatory targets”. To avoid confusion, we now used the phrase “bound genes” and “ChIP-seq targets” in the revised manuscript.

Reviewer #2 (Remarks to the Author):

The authors have made a genuinely impressive effort to improve the manuscript, with a large body of additional analysis and experimentation. The manuscript is much clearer in terms of writing, and the figures are excellent. I am pleased that the authors took the critique on board and made an extensive effort to produce a manuscript that is more compelling, engaging and informative. Well done.

My previous concerns (and more) have been addressed to my satisfaction, in particular the cross-species complementation experiment and the analysis that reveals the importance of TF co-binding. I only have very minor comments.

Abstract: although generally the writing has been significantly improved, I think the abstract could be refined a little more for clarity. For example, lines 34-36 could be improved thus (just a suggestion, I appreciate word limits may be a problem): "GLK-dependent gene expression is associated with highly conserved binding sites, but many sites near genes that are not differentially expressed in *glk* mutants are nevertheless under purifying selection. Accordingly, we trained machine learning models to predict the transcriptional outcome of GLK binding sites using multiple..."

Thanks for the suggestion. We modified the abstract to:

The conserved binding sites are often found near genes dependent on GLK for expression, but many sites near genes that are not differentially expressed in the *glk* mutant are nevertheless under purifying selection. Accordingly, we trained machine learning models to predict the transcriptional regulatory function of GLK binding sites using multiple

Line 66: what are "given conditions"?

We have shortened the sentence to "...found that many TF binding sites have no transcriptional effect".

Line 432, lyrata

Corrected. Thanks for spotting that.

Reviewer #3 (Remarks to the Author):

I found that the authors adequately and appropriately addressed the concerns I brought up in my initial review. I feel that I cannot comment on the issues raised by the other reviewers, however. I think the work and the conclusions presented in the manuscript are important even given the current limitations of analysis. The authors have recognized most of the

limitations of their analysis and the manuscript provides an important road map for future work that will will expand the interpretations on TF function and cis regulatory element roles. Aside from a small typographic error in line 215, I did not find obvious errors. The manuscript has been improved considerably in response to the reviewers' comments.

Reviewer #4 (Remarks to the Author):

The revised manuscript resolves all issues I had in the original review. The new analysis, figures and text add valuable new information. Altogether the manuscript provides a very thorough analysis of GLK cistrome conservation and variation, and the impact on gene expression. In particular, the GLK mutant expression analysis of the two factors is powerful and illustrative.

Reviewers' Comments:

Reviewer #1:

Remarks to the Author:

I am satisfied with the latest explanations provided by the authors. Congratulations on this very interesting work and set of analyses.